# Decoding stringy near-supersymmetric black holes

Chi-Ming Chang[1,2*], Li Feng[1,3,4†], Ying-Hsuan Lin[5‡] and Yi-Xiao Tao[1,6∘]

**1** Yau Mathematical Sciences Center (YMSC), Tsinghua University, Beijing 100084, China
**2** Beijing Institute of Mathematical Sciences and Applications (BIMSA),
Beijing 101408, China
**3** International Centre for Theoretical Physics Asia-Pacific,
University of Chinese Academy of Sciences, 100190 Beijing, China
**4** School of Physics and Technology, Wuhan University, Wuhan, Hubei 430072, China
**5** Jefferson Physical Laboratory, Harvard University, Cambridge, MA 02138, USA
**6** Department of Mathematical Sciences, Tsinghua University, Beijing 100084, China

⋆ cmchang@tsinghua.edu.cn , † lifeng.phys@gmail.com ,
‡ yhlin@fas.harvard.edu , ∘ taoyx21@mails.tsinghua.edu.cn

## Abstract

Building on the recent discovery of the first candidate black hole operator in the $\mathcal{N} = 4$ super-Yang-Mills, we explore the near-supersymmetric aspects of the theory that capture lightly excited, highly stringy black holes. We extend the superspace formalism describing the classically supersymmetric (1/16-BPS) sector of $\mathcal{N} = 4$ super-Yang-Mills and compute a large number of one-loop anomalous dimensions. Despite being in the highly stringy regime, we find hints of a gap in the spectrum, similar to that found by a gravitational path integral. We also determine the actual expression of the first candidate black hole operator at weak gauge coupling, going beyond the cohomological construction.

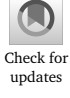

# 1  Introduction and summary

Since the discovery of the emblematic holographic duality between type IIB string theory and $\mathcal{N} = 4$ super-Yang-Mills (SYM) twenty-five years ago, various remarkable connections and matches have been established. Keys to bridging the two far-apart regimes of simplicity include integrability in the planar limit [1], supersymmetry protection [2], conformal bootstrap [3], and large charge universality [4] (see also the references contained in these reviews). Although non-planar non-supersymmetric aspects of the theory are computable in each of the dual descriptions, without an overlapping regime of validity, the statement of a duality covering all corners of the theory becomes unverifiable.

The advent of the holographic duality opened a new chapter in the study of black holes and surrounding puzzles. Black holes' existence has long necessitated a deeper understanding of the quantum nature of gravity, as the singularity signals a breakdown of general relativity, and the No-Hair theorem prevents any classical explanation of the statistical entropy. In the holographic setting, the gauge theory supplies a non-perturbative description of a bulk black hole as an ensemble of microstates. Reproducing the salient features of black holes, such as their entropy, within the gauge theory framework is widely regarded as a highly significant test of the duality, as well as an affirmation of string theory as a consistent theory of quantum gravity. Moreover, exact *supersymmetric* results coming from gauge-theoretic computations have supplied key guidance in bettering our first-principle understanding of the gravitational path integral.

Until lately, the main focus on the $\mathcal{N} = 4$ SYM side has been the study of its index, with a triumph being the successful match of the asymptotic growth of the (signed) degeneracies with the black hole entropy [5–7]. However, the index counts all states equally, blending gravitons, black holes, and whatnot, as well as all their bound states into a set of numbers. One can

embrace more refined studies of supersymmetric states, going beyond indiscriminate counting. Pursuit in this direction was attempted by [8–13], and achieved a recent breakthrough by [14] in constructing a first candidate black hole (non-graviton) microstate (in the cohomological sense). Generalizations and further progress were made in [15–17].

Let us quickly review this line of development. By the standard argument in Hodge theory, there is a one-to-one correspondence between the supersymmetric states and the cohomology of the associated supercharge $Q$ [11]. The cohomologies corresponding to gravitons are fully recognized and can be completely separated from the rest [13]. It has been conjectured [11] and perturbatively proven [14] that away from the free point $g_{YM} = 0$, the spectrum of supersymmetric states (captured by the dimensions of the $Q$-cohomologies) does not depend on the $g_{YM}$. In [14], an extensive enumeration of cohomology classes at weak coupling, extending to high energies and charge ranges, unveiled the first non-graviton cohomology for gauge group SU(2).

A separate line of development in recent years is the study of "coarse-grained" holographic dualities between simple gravitational theories and ensemble-averaged (quench-disordered) non-gravitational systems and has achieved remarkable success. While the philosophical nature of such dualities is a subject of intense debate, much less controversial is the embedding of these coarse-grained dualities inside "fine-grained" dualities as effective descriptions. In a series of papers [18–22], the dynamics near an extremal black hole's horizon, which contains an AdS$_2$ factor, has been analyzed using an effective Jackiw-Teitelboim (JT) gravitational path integral. For near-supersymmetric black holes, their main result is a quantitative prediction of the near-BPS spectrum. These developments on the gravitational side pose some natural questions to the gauge-theoretic framework. To name a few: Can the effective JT supergravity description be deduced within the gauge-theoretic framework? Are there *qualitative* features shared between (or perhaps interpolating) the weak and strong coupling regimes?

**Summary of this work**   This paper develops tools for studying $\mathcal{N} = 4$ SYM in the non-planar near-supersymmetric regime.

- Expanding the superspace formalism of [13,14], we drastically simplify the Hamiltonian encoding the spectrum of one-loop anomalous dimensions in a subsector of the full theory comprising operators that are BPS in the free $g_{YM} \to 0$ limit. We also clarify the relationship among operators at different couplings, and in what sense near-supersymmetric black holes are captured by this subsector.

- We systematically construct and diagonalize the Hamiltonian across a large range of charges. The resulting near-BPS spectrum at weak coupling shows features reminiscent of the spectrum at strong 't Hooft coupling, which is captured by $\mathcal{N} = 2$ JT supergravity [20]; in particular, there are hints of a "gap" (a proper notion of which requires large $N$).

- Diagonalizing the Hamiltonian in the charge sector of the first non-graviton cohomology gives the precise weak-coupling form of the candidate black hole operator (a specific representative of the cohomology), which was also computed in simultaneous work [23].

The remainder of this paper is organized as follows. Section 2 introduces the classically-BPS sector of $\mathcal{N} = 4$ SYM and discusses its Hilbert space and Hamiltonian. Section 2.3 discusses the near-horizon excitation of near-BPS black holes and argues a conjecture on the relation between them and the classically-BPS sector. Section 3 develops the matrix representations for the supercharge $Q$ and the one-loop dilatation operator $H$. Section 4 give differential representations for $Q$ and $H$. Section 5 presents our results and gives discusses the implications and outlooks.

## 2 Classically-BPS sector and near-BPS black holes

This section characterizes the classically-BPS sector of $\mathcal{N} = 4$ super-Yang-Mills (SYM), describes its symmetry algebra, and explains how it captures near-BPS physics even in the strong coupling regime.

### 2.1 Hilbert space and level repulsion

The $\mathcal{N} = 4$ super-Yang-Mills is a supersymmetric gauge theory with two marginal parameters, a gauge coupling $g_{\text{YM}}$ and a topological angle $\theta$. The fields and action are detailed in Appendix D. To define the classically-BPS sector describing near-BPS excitations, let us begin with the Hilbert space of local operators in $\mathcal{N} = 4$ SYM. The $\mathfrak{psu}(2, 2|4)$ superconformal algebra, which is reviewed in Appendix B, acts on the Hilbert space as linear maps. To define BPS states, we pick a supercharge and its Hermitian conjugate (BPZ conjugate)

$$Q \equiv Q_-^4, \qquad Q^\dagger = S_4^-, \tag{1}$$

whose anti-commutator is

$$\Delta \equiv 2\{Q, Q^\dagger\} = D - 2J_L - q_1 - q_2 - q_3. \tag{2}$$

Unitarity implies that all the states or operators must satisfy the BPS bound

$$\Delta \geq 0. \tag{3}$$

The states that saturate the BPS bound (3) are called BPS states.[1]

In perturbation theory, the dilation operator $D$ admits an expansion

$$D = D^{(0)} + g_{\text{YM}}^2 D^{(2)} + g_{\text{YM}}^4 D^{(4)} + \cdots. \tag{4}$$

The leading term $D^{(0)}$ is the classical (bare) dimension. The *classically-BPS states* are those that satisfy the classical BPS condition

$$\Delta^{(0)} \equiv D^{(0)} - 2J_L - q_1 - q_2 - q_3 = 0. \tag{5}$$

The classically-BPS sector is the space that contains all the classically-BPS states. It is a closed sector because under operator-mixing the classically-BPS operators do not mix with classically non-BPS operators. This fact can be argued in perturbation theory by noting that the classical dimension commutes with all the higher loop dilatation operators [24], i.e. $[D^{(0)}, D^{(n)}] = 0$. Hence, operator-mixing only occurs among operators with the same angular momenta, R-charges, *and* classical dimension. Even at finite Yang-Mills coupling $g_{\text{YM}}$, one still expects the classically-BPS sector to be well-defined. The classical dimension of an operator can be defined *adiabatically*, by following its energy (conformal dimension) along a path with no level crossing to weak coupling.

By the von Neumann-Wigner theorem (level repulsion), level crossing in the operator spectrum only occurs at real codimension-two submanifolds of the conformal manifold. The conformal manifold of $\mathcal{N} = 4$ SYM is a complex one-dimensional space, parametrized by the complexified gauge coupling

$$\tau = \frac{\theta}{2\pi} + \frac{4\pi i}{g_{\text{YM}}^2}. \tag{6}$$

---

[1]The states that do not saturate (3) but saturate the BPS bound of other supercharges would not be referred to as BPS states in this paper.

Hence, level crossings only happen at isolated points on the complex $\tau$-plane, and we define the classical dimension of an operator at generic coupling $\tau$ by following a path to $\tau = i\infty$ that avoids all these points.[2]

The anomalous dimension $D - D^{(0)}$ in the classically-BPS sector can be naturally regarded as the Hamiltonian of a supersymmetric quantum mechanics,

$$H \equiv D - D^{(0)} = 2\{Q, Q^\dagger\}. \tag{7}$$

The perturbative expansion of $H$ starts at one-loop order. As argued in [14], with a suitable regularization scheme, the supercharge $Q$ receives no quantum corrections perturbatively. Therefore, on the right-hand side of (7), the perturbative corrections must be encoded in the Hermitian conjugation (BPZ conjugation) $\dagger$. We will focus on the one-loop Hamiltonian, where $\dagger$ is induced by the inner product in the free theory ($g_{\text{YM}} = 0$).[3] In the following, $H$ always refers to the one-loop Hamiltonian, and $\dagger$ to the free Hermitian conjugation.

## 2.2 Symmetry algebra and characters

The symmetry algebra of the classically-BPS sector is the centralizer of $\Delta$ in $\mathfrak{psu}(2,2|4)$, which can be written as a semi-direct product

$$\mathcal{C}(\Delta) = \mathfrak{u}(1) \ltimes [\mathfrak{psu}(1,2|3) \times \mathfrak{su}(1|1)], \tag{8}$$

and whose generators are detailed in Appendix C. Let us define the $\mathcal{C}(\Delta)$ primary operators to be the operators annihilated by all the superconformal supercharges inside $\mathcal{C}(\Delta)$. The superconformal supercharges outside $\mathcal{C}(\Delta)$ all have $\Delta < 0$ except $S_4^+$ which has $\Delta = 2$. If a $\mathcal{C}(\Delta)$ primary operator $\mathcal{O}$ is also annihilated by $S_4^+$, then it is also a $\mathfrak{psu}(2,2|4)$ superconformal primary, otherwise it is a descendent of the $\mathfrak{psu}(2,2|4)$ superconformal primary $S_4^+(\mathcal{O})$. If $\mathcal{O}$ is a BPS operator, it is a superconformal primary in a **bx**-type superconformal multiplet when $J_L = 0$, and it is a superconformal descendent in a **cx**-type superconformal multiplet when $J_L > 0$ [8]. If $\mathcal{O}$ is a non-BPS operator, by the decomposition rules (2.16) in [8], $\mathcal{O}$ should fall into a **cx**-type superconformal multiplet in the free limit $g_{\text{YM}} \to 0$, and hence should be a superconformal descendent.

The spectrum of the classically-BPS sector can be summarized by a partition function as

$$Z(\beta, \omega_i, \Phi_i) = \text{Tr}_{\Delta^{(0)}=0} \left( e^{-\beta\Delta - J_1\omega_1 - J_2\omega_2 - q_1\Phi_1 - q_2\Phi_2 - q_3\Phi_3} \right). \tag{9}$$

The partition function can be organized by the centralizer $\mathcal{C}(\Delta)$ symmetry. We split the partition function into a sum of the BPS and non-BPS partition functions as

$$Z(\beta, \omega_i, \Phi_i) = Z_{\text{BPS}}(\beta, \omega_i, \Phi_i) + Z_{\text{non-BPS}}(\beta, \omega_i, \Phi_i), \tag{10}$$

where $Z_{\text{BPS}}$ contains the contributions from the BPS ($\Delta = 0$) states, and $Z_{\text{non-BPS}}$ contains the contributions from the non-BPS states in the classically-BPS states.

We will focus on the non-BPS partition function $Z_{\text{non-BPS}}(\beta, \omega_i, \Phi_i)$. The non-BPS states in the classically-BPS sector form long multiplets of the centralizer $\mathcal{C}(\Delta)$. Hence,

---

[2]It is worth mentioning that level crossings are only expected to happen at the free points, i.e. the $\text{PSL}(2,\mathbb{Z})$ images of $\tau = i\infty$, and the level crossings in the planar limit at finite 't Hooft coupling disappear after including $1/N$ corrections [25].

[3]In our convention specified in Appendix D, $g_{\text{YM}}$ is an overall coupling constant of the action (D.9). Hence, the two-point functions (inner products) of the fundamental fields in free theory are all proportional to $g_{\text{YM}}^2$. Alternatively, one could rescale the fundamental fields to normalize their two-point functions. This would make the supercharge $Q$ of order $g_{\text{YM}}$ when acting on classically-BPS operators.

$Z_{\text{non-BPS}}(\beta, \omega_i, \Phi_i)$ can be decomposed into characters $\chi_{\Delta, J_L, J_R, q_i}(\beta, \omega_i, \Phi_i)$ of the long multiplets,

$$
\begin{aligned}
\chi_{\Delta, J_L, J_R, q_i}(\beta, \omega_i, \Phi_i) = {} & \frac{e^{-\beta\Delta - J_L(\omega_1+\omega_2) - \frac{1}{3}(\Phi_1+\Phi_2+\Phi_3)(q_1+q_2+q_3)}\chi_{J_R}(e^{-\omega_1+\omega_2})}{(1-e^{-\omega_1})(1-e^{-\omega_2})} \\
& \times \chi_{q_1-q_2, q_2-q_3}\left(e^{-\frac{1}{3}(2\Phi_1-\Phi_2-\Phi_3)}, e^{-\frac{1}{3}(\Phi_1+\Phi_2-2\Phi_3)}\right) \\
& \times \prod_{\substack{t_i, s_i = \pm 1 \\ t_1+t_2+s_1+s_2+s_3=1}}\left(1 + e^{-\frac{1}{2}(t_1\omega_1+t_2\omega_2+s_1\Phi_1+s_2\Phi_2+s_3\Phi_3)}\right),
\end{aligned}
\tag{11}
$$

where $\chi_{J_R}$ and $\chi_{q_1-q_2, q_2-q_3}$ are the $\mathfrak{su}(2)$ and $\mathfrak{su}(3)$ characters, explicitly given by

$$
\chi_{J_R}(e^{-\omega}) = \sum_{n=0}^{2J_R} e^{(n-J_R)\omega},
$$

$$
\chi_{\mathcal{R}_1, \mathcal{R}_2}(e^{-\Omega_1}, e^{-\Omega_2}) = e^{-(\mathcal{R}_1+2\mathcal{R}_2)\Omega_1}\sum_{k=\mathcal{R}_2}^{\mathcal{R}_1+\mathcal{R}_2}\sum_{l=0}^{\mathcal{R}_2}e^{\frac{3(k+l)\Omega_1}{2}}\frac{\sinh\frac{(k-l+1)(2\Omega_2-\Omega_1)}{2}}{\sinh\frac{2\Omega_2-\Omega_1}{2}},
\tag{12}
$$

where the $\mathfrak{su}(3)$ characters are labeled by the Dynkin labels $\mathcal{R}_1 = \mathcal{R}_1^1 - \mathcal{R}_2^2 = q_1 - q_2$ and $\mathcal{R}_2 = \mathcal{R}_2^2 - \mathcal{R}_3^3 = q_2 - q_3$.

## 2.3 Near-BPS black holes

Under the AdS/CFT correspondence, in the 't Hooft large $N$ limit, generic BPS states with large angular momenta and R-charges of order

$$
J_1, J_2, q_1, q_2, q_3 \sim N^2 \gg 1,
\tag{13}
$$

are dual to the microstates of 1/16-BPS black holes in AdS$_5 \times$ S$_5$ [26–30]. Here $J_1$, $J_2$ are rotation generators along the two-planes in $\mathbb{R}^4$, and are related to $J_L$, $J_R$ by (B.9). For a black hole to have a macroscopic event horizon, all five angular momenta and R-charges (13) must be activated. Furthermore, standard BPS black hole solutions in AdS$_5$ are subject to a charge relation, see e.g. (2.85) in [6]. However, as discussed in [31], "revolving black holes"—standard BPS black hole solutions boosted by momenta $P$—easily violate the charge relation (in specific one direction). From a gauge-theoretic standpoint, the charge relation does not seem natural: at finite $N$, the quantization of charges makes it impossible to satisfy the relation exactly.

Excitations of black holes near the BPS bound have been studied in [20]. Let us recap some key results. The near-horizon geometry of a BPS black hole has an AdS$_2$ factor and develops an emergent local $\mathfrak{su}(1, 1|1)$ superconformal algebra. The finite-temperature quantum corrections break this local symmetry to a global $\mathfrak{su}(1, 1|1)$, which acts as the isometry of the near-horizon region. The dimensional reduction of the ten-dimensional IIB supergravity down to AdS$_2$ is expected to produce an $\mathcal{N} = 2$ Jackiw-Teitelboim (JT) supergravity, whose boundary $\mathcal{N} = 2$ Schwarzian theory captures the Goldstone modes of the symmetry breaking. A distinguishing feature of the $\mathcal{N} = 2$ Schwarzian theory from its less supersymmetric cousins is the presence of a gap in its spectrum above the ground states [32,33]. This implies that the low-lying spectrum of BPS and near-BPS black holes consists of order $e^{N^2}$ isolated BPS states and a continuum of near-BPS states with an order $N^{-2}$ gap separating them from the BPS states. More precisely, in a sector with large fixed angular momenta and R-charges (13), the near-BPS states in the continuum have dimensions above the bound

$$
\Delta > \Delta_{\text{BPS}} + \Delta_{\text{gap}}, \quad \Delta_{\text{gap}} = \frac{\widetilde{\Delta}}{N^2},
\tag{14}
$$

where $\widetilde{\Delta}$ is an explicit function of the angular momenta and R-charges, given in (3.91) of [20].

Let us argue that the gap, together with the lower end of the continuum in the spectrum, is captured by the classically-BPS sector in the $\mathcal{N} = 4$ SYM. In the weak coupling limit, the classically non-BPS states have $\Delta \gtrsim \frac{1}{2}$ because $\Delta^{(0)} \geq \frac{1}{2}$. By the von Neumann-Wigner theorem (level repulsion), when going from weak to strong coupling, the $\Delta$'s of the classically non-BPS states cannot become lower than those of the classically-BPS states. Hence, the states at the lower end of the continuum must be classically-BPS.

In the next section, we develop a matrix representation for $H$ and diagonalize it for small ranks $N = 2, 3, 4$ up to relatively high angular momenta and R-charges. Our results lead us to further conjecture that the gap $\Delta_{\text{gap}}$ is a continuous function of the 't Hooft coupling $\lambda = g_{\text{YM}}^2 N$, behaving as

$$\Delta_{\text{gap}} = \frac{\widetilde{\Delta}(\lambda)}{N^2}, \quad \widetilde{\Delta} = \widetilde{\Delta}^{(1)}\lambda + \mathcal{O}(\lambda^2), \tag{15}$$

in the weak coupling limit and charge regime (13). In other words, the spectrum of the Hamiltonian $H$ in (7) has a gap of order $g_{\text{YM}}^2/N$ in the sector with large angular momenta and R-charges (13).

# 3 Matrix representation

In this section, we review the superspace formalism of [13], introduce an explicit basis for the classically-BPS Hilbert space of $\mathcal{N} = 4$ super-Yang-Mills (SYM), derive a compact formula for the inner product, and compute the matrix representations of the supercharge $Q$ and the one-loop Hamiltonian $H$.

## 3.1 Superspace formalism

In perturbation theory, the classically-BPS operators can be constructed by gauge-invariant combinations of fundamental fields and covariant derivatives that classically saturate the BPS bound (see Appendix D)

$$\phi^i \equiv \Phi^{4i}, \quad \psi_i \equiv -i\Psi_{+i}, \quad \lambda_{\dot{\alpha}} \equiv \overline{\Psi}_{\dot{\alpha}}^4, \quad f = -iF_{++} \equiv i(\sigma^{\mu\nu})_{++}F_{\mu\nu}, \quad D_{\dot{\alpha}} \equiv D_{+\dot{\alpha}}, \tag{16}$$

for $i = 1, 2, 3$ and $\dot{\alpha} = \dot{+}, \dot{-}$. They are referred to as BPS *letters*. The BPS letters can be assembled nicely into a superfield $\Psi(z^{\dot{\alpha}}, \theta_i)$ in the superspace $\mathbb{C}^{2|3}$ as [13]

$$\Psi(z^{\dot{\alpha}}, \theta_i) = -i\sum_{n=0}^{\infty} \frac{1}{n!}(z^{\dot{\alpha}}D_{\dot{\alpha}})^n \left[\frac{1}{n+1}z^{\dot{\beta}}\lambda_{\dot{\beta}} + 2\theta_i\phi^i + \epsilon^{ijk}\theta_i\theta_j\psi_k + 4\theta_1\theta_2\theta_3 f\right], \tag{17}$$

where $z^{\dot{\alpha}}$ and $\theta_i$ are the bosonic and fermionic superspace coordinates. The supercharge $Q$ acts on the superfield $\Psi$ as

$$\{Q, \Psi\} = \Psi^2, \tag{18}$$

and obeys the Leibniz rule when acting on composites of $\Psi$'s. The BPS letters can be recovered by taking superspace derivatives as

$$\Psi^A \equiv \partial_{z^+}^{a_1}\partial_{z^-}^{a_2}\partial_{\theta_1}^{a_3}\partial_{\theta_2}^{a_4}\partial_{\theta_3}^{a_5}\Psi(z, \theta)\big|_{z=0, \theta=0}, \tag{19}$$

where $A = (a_1, \cdots, a_5)$. Using the trace basis, the classically-BPS sector is spanned by the multi-traces

$$\text{tr}(\Psi^{A_1}\cdots\Psi^{A_m})\text{tr}(\Psi^{B_1}\cdots\Psi^{B_n})\cdots, \tag{20}$$

which would be referred to as the BPS *words*. The multi-traces are subject to trace relations, which can be eliminated by substituting explicit $N \times N$ (traceless) matrices for $\Psi^A$.

## 3.2 Basis and inner product

Consider a subspace $V$ of the classically-BPS sector spanned by the BPS words with a fixed number $(n_{z^+}, n_{z^-}, n_{\theta_1}, n_{\theta_2}, n_{\theta_3})$ of derivatives $\partial_{z^+}, \partial_{z^-}, \partial_{\theta_1}, \partial_{\theta_2}, \partial_{\theta_3}$. Such BPS words, denoted by $w_i$, can be assembled into a finite-dimensional row vector $\vec{w} = (w_1, w_2, \cdots)$. After substituting explicit matrices, we can expand the BPS words $w_i$ in terms of the monomials $t_i$ of the matrix components, whereas, for traceless matrices, we use the traceless condition to substitute the $(N, N)$-component of the matrix. The expansion can be written explicitly as

$$\vec{w} = \vec{t}\,\mathbf{A}', \tag{21}$$

where $\vec{t} = (t_1, t_2, \cdots)$ is a row vector of monomials $t_i$ and $\mathbf{A}'$ is the matrix of the coefficients of the expansion. Now, we can eliminate the trace relations between $w_i$ by column reducing the matrix $\mathbf{A}'$. Let us denote the column reduced matrix by $\mathbf{A}$. A complete basis of the subspace $V$ is given by the elements of the row vector $\vec{\mathcal{O}} = (\mathcal{O}_1, \mathcal{O}_2, \cdots)$,

$$\vec{\mathcal{O}} = \vec{t}\,\mathbf{A}. \tag{22}$$

Let $|\mathcal{O}_i\rangle$ be the state corresponding to the operators $\mathcal{O}_i$. We denote the inner product matrix of the states $|\mathcal{O}_i\rangle$'s by $\mathbf{M}$,

$$\mathbf{M}_{ij} \equiv \langle \mathcal{O}_i | \mathcal{O}_j \rangle, \quad \mathbf{M}^\dagger = \mathbf{M}. \tag{23}$$

$\mathbf{M}$ is related to the inner product matrix $\mathbf{T}_{ij} = \langle t_i | t_j \rangle$ of the monomials $t_i$'s by

$$\mathbf{M}_{ij} = \langle \mathcal{O}_i | \mathcal{O}_j \rangle = \sum_{k,l} \mathbf{A}^*_{ki} \mathbf{A}_{lj} \langle t_k | t_l \rangle = (\mathbf{A}^\dagger \mathbf{T} \mathbf{A})_{ij}. \tag{24}$$

As discussed in Section 2.1, since we focus on the one-loop Hamiltonian, we use the inner product $\mathbf{T}$ in the free theory, which can be computed by the two-point functions,

$$\left\langle t_i(x)^\dagger t_j(0) \right\rangle = \frac{\langle t_i | t_j \rangle}{|x|^{2\Delta_0}}, \tag{25}$$

where $t_i$ and $t_j$ have the same classical dimension $\Delta_0$, otherwise the two-point function vanishes.

In the free theory, the two-point function can be simply computed by Wick contractions. More explicitly, consider a typical monomial

$$(\Psi^{A_1})^{I_1}_{J_1} \cdots (\Psi^{A_Y})^{I_Y}_{J_Y}, \tag{26}$$

where the upper (or lower) $I, J = 1, \cdots, N$ indices are the SU($N$) or U($N$) fundamental (or antifundamental) indices. The inner product of the monomial (26) with itself factorizes as

$$\left\langle (\Psi^{A_n})^{I_n}_{J_n} \cdots (\Psi^{A_1})^{I_1}_{J_1} \middle| (\Psi^{A_1})^{I_1}_{J_1} \cdots (\Psi^{A_Y})^{I_Y}_{J_Y} \right\rangle$$
$$= \sum_{\pi \in S_Y} (-1)^{N_\pi} \left\langle (\Psi^{A_1})^{I_1}_{J_1} \middle| (\Psi^{A_{\pi(1)}})^{I_{\pi(1)}}_{J_{\pi(1)}} \right\rangle \cdots \left\langle (\Psi^{A_n})^{I_n}_{J_n} \middle| (\Psi^{A_{\pi(Y)}})^{I_{\pi(Y)}}_{J_{\pi(Y)}} \right\rangle, \tag{27}$$

where the sum is over all the permutations $\pi \in S_Y$, and $N_\pi$ is the number of commutations between fermionic letters.

In Appendix E, we explicitly compute the inner product matrix of single letters in the superfield basis, and find a rather compact result

$$\left\langle \partial^{a_1}_{z^+} \partial^{a_2}_{z^-} \partial^{a_3}_{\theta_1} \partial^{a_4}_{\theta_2} \partial^{a_5}_{\theta_3} \Psi^J_I \middle| \partial^{a_1}_{z^+} \partial^{a_2}_{z^-} \partial^{a_3}_{\theta_1} \partial^{a_4}_{\theta_2} \partial^{a_5}_{\theta_3} \Psi^L_K \right\rangle$$
$$= \frac{g^2_{YM}}{2^{4+2a_1+2a_2}\pi^2} \Gamma(a_1+1)\Gamma(a_2+1)\Gamma(a_1+a_2+a_3+a_4+a_5)$$
$$\times \begin{cases} \delta^J_K \delta^L_I, & \text{for} \quad \text{U}(N), \\ \delta^J_K \delta^L_I - \frac{1}{N} \delta^J_I \delta^L_K, & \text{for} \quad \text{SU}(N). \end{cases} \tag{28}$$

### 3.3 Matrix representation for $Q$ and $H$

The supercharge $Q$ action (18) does not change the number of derivatives, and hence maps the space $V$ to itself. Let $Y$ be the number of $\Psi$'s in a BPS word, and $V_Y$ be the subspace of $V$ with a fixed $Y$. $Q$ acts on $V$ as a chain complex

$$\cdots \xrightarrow{Q} V_{Y-1} \xrightarrow{Q} V_Y \xrightarrow{Q} V_{Y+1} \xrightarrow{Q} \cdots. \tag{29}$$

We will write $\vec{\mathcal{O}}_Y$, $\vec{t}_Y$ and $\mathbf{A}_Y$ for those with a fixed number of $\Psi$'s. By acting the supercharge on the vector $\vec{t}_Y$ and then re-expanding the result in terms of the elements in $\vec{t}_{Y+1}$, we find

$$Q(\vec{t}_Y) = \vec{t}_{Y+1} \mathbf{q}_Y, \tag{30}$$

where $q$ is a matrix of the expansion coefficients. Substituting it into (22), we find

$$Q(\vec{\mathcal{O}}_Y) = Q(\vec{t}_Y)\mathbf{A}_Y = \vec{t}_{Y+1}\mathbf{q}_Y\mathbf{A}_Y. \tag{31}$$

The dimension of the $Q$-cohomology is given by the rank of the matrices as

$$\dim(V_Y) - \dim(QV_Y) - \dim(QV_{Y-1})$$
$$= \operatorname{rank}(\mathbf{A}_Y) - \operatorname{rank}(\mathbf{q}_Y\mathbf{A}_Y) - \operatorname{rank}(\mathbf{q}_{Y-1}\mathbf{A}_{Y-1}). \tag{32}$$

Consider a pair of bra and ket states, $\langle \mathcal{O}_i |$ and $| \mathcal{O}_j \rangle$, with $Y+1$ and $Y$ numbers of $\Psi$'s, respectively. Sandwiching the supercharge $Q$ between them, we obtain a matrix $\mathbf{Q}_Y$ as

$$(\mathbf{Q}_Y)_{ij} \equiv \langle \mathcal{O}_i | Q | \mathcal{O}_j \rangle = \sum_{k,l} (\mathbf{A}_{Y+1})^*_{ki} (\mathbf{q}_Y\mathbf{A}_Y)_{lj} \langle t_k | t_l \rangle = (\mathbf{A}^\dagger_{Y+1}\mathbf{T}_{Y+1}\mathbf{q}_Y\mathbf{A}_Y)_{ij}, \tag{33}$$

where the states $\langle t_k |$ and $| t_l \rangle$ each have $Y+1$ number of $\Psi$'s.

Now, sandwiching the Hamiltonian $H$ between a pair of bra and ket states, $\langle \mathcal{O}_i |$ and $| \mathcal{O}_j \rangle$, each having $Y$ numbers of $\Psi$'s, we obtain a matrix $\mathbf{H}_Y$ as

$$(\mathbf{H}_Y)_{ij} \equiv \langle \mathcal{O}_i | H | \mathcal{O}_j \rangle. \tag{34}$$

Using the commutator (7), we find

$$(\mathbf{H}_Y)_{ij} = 2\left( \langle \mathcal{O}_i | QQ^\dagger | \mathcal{O}_j \rangle + \langle \mathcal{O}_i | Q^\dagger Q | \mathcal{O}_j \rangle \right)$$
$$= 2\left( \mathbf{Q}_{Y-1}\mathbf{M}^{-1}_{Y-1}\mathbf{Q}^\dagger_{Y-1} + \mathbf{Q}^\dagger_Y\mathbf{M}^{-1}_{Y+1}\mathbf{Q}_Y \right)_{ij}, \tag{35}$$

where the matrix $\mathbf{M}_Y$ is the inner product matrix in the subspace $V_Y$, given by restricting the matrix (24) as

$$\mathbf{M}_Y = \mathbf{A}^\dagger_Y\mathbf{T}_Y\mathbf{A}_Y. \tag{36}$$

Because our basis $| \mathcal{O}_i \rangle$ is not orthonormal, the eigenvalues of the Hamiltonian are not the eigenvalues of $\mathbf{H}_Y$, but instead the eigenvalues of the matrix $\mathbf{h}_Y$ given by $H$ acting on the basis vector $| \mathcal{O}_i \rangle$ (with $Y$ number of $\Psi$'s),

$$H | \mathcal{O}_i \rangle \equiv \sum_j | \mathcal{O}_j \rangle (\mathbf{h}_Y)_{ji}. \tag{37}$$

The matrices $\mathbf{h}_Y$ and $\mathbf{H}_Y$ are related by

$$\mathbf{h}_Y = \mathbf{M}^{-1}_Y\mathbf{H}_Y. \tag{38}$$

We have obtained all the ingredients for computing the $\mathbf{h}_Y$. The inner product matrix $\mathbf{T}_Y$ was computed in the previous subsection and Appendix E. The matrices $\mathbf{A}_Y$ and $\mathbf{q}_Y$ are

computed for small ranks $N = 2, 3, 4$ and a large class of different $(n_{z^+}, n_{z^-}, n_{\theta_1}, n_{\theta_2}, n_{\theta_3})$'s and $Y$'s in [14].

Finally, let us comment on the construction of the graviton cohomology, which is defined as the cohomology represented by the operators in (5.1) in [13]. Let $\vec{g}_Y$ be a row vector of these representatives in the trace form. Similar to what we did around (21), by substituting the explicit matrices for each BPS letter and expanding the result, we get

$$\vec{g}_Y = \vec{t}_Y \mathbf{B}'_Y \,, \tag{39}$$

where $\mathbf{B}'_Y$ is the matrix of the coefficients of the expansion. Next, we column reduce the matrix $\mathbf{B}'_Y$ getting a matrix $\mathbf{B}_Y$. The independent representatives of the graviton cohomology are contained in the row vector

$$\vec{\mathcal{G}}_Y = \vec{t}_Y \mathbf{B}_Y \,. \tag{40}$$

The dimension of the graviton cohomology can be computed by

$$\mathrm{rk}(\mathbf{B}_Y, \mathbf{q}_{Y-1}\mathbf{A}_{Y-1}) - \mathrm{rk}(\mathbf{q}_{Y-1}\mathbf{A}_{Y-1}) \,, \tag{41}$$

where the matrix $(\mathbf{B}_Y, \mathbf{q}_{Y-1}\mathbf{A}_{Y-1})$ is given by concatenating the matrices $\mathbf{B}_Y$ and $\mathbf{q}_{Y-1}\mathbf{A}_{Y-1}$. To know whether a given BPS operator $\mathcal{O}$ is a (multi-)graviton operator, we could check that if $\mathcal{O}$ is in the same cohomology class as the operators in the row vector $\vec{\mathcal{G}}_Y$.

## 3.4 Large $N$ limit

In the large $N$ limit with fixed charges, the standard 't Hooft's argument tells us that we can focus on the single-trace operators since the anomalous dimensions of multi-trace operators are given by the sums of those of the single-trace constituents.[4] Let us see this explicitly in our formalism.

We start with the vector $\vec{w}$ of BPS words defined in Section 3.2, and further order the entries of $\vec{w}$ according to the number of traces. Using the inner product of the BPS letter (28), we compute the inner product matrix

$$\mathcal{M}_{ij} \equiv \langle w_i | w_j \rangle \,, \quad \mathcal{M}^\dagger = \mathcal{M} \,. \tag{42}$$

At finite $N$, the matrix $\mathcal{M}$ is degenerate with the null space spanned by the trace relations. In the large $N$ limit, there is no trace relation, and the words with different numbers of traces have inner products suppressed by $N^{-1}$.[5] Hence, at the leading order, the inner product matrix $\mathcal{M}$ is nondegenerate and block diagonal. One can further argue that the block $\mathcal{M}_n$ associated with the $n$-trace words is the $n$-th tensor power of the block associated with single-trace words, i.e.

$$\mathcal{M}_n = \underbrace{\mathcal{M}_1 \otimes \cdots \otimes \mathcal{M}_1}_{n} \,. \tag{43}$$

The action of the supercharge $Q$ on the BPS words $\vec{w}$ gives a matrix $\mathcal{Q}$ as

$$Q(\vec{w}) = \vec{w}\,\mathcal{Q} \,. \tag{44}$$

As the $Q$-action does not change the number of traces, the matrix $\mathcal{Q}$ is also block diagonal. Furthermore, by the Leibniz rule of the $Q$-action, the block $\mathcal{Q}_n$ of the $n$-trace words should take the form

$$\mathcal{Q}_n = \mathcal{Q}_1 \otimes \underbrace{I \otimes \cdots \otimes I}_{n-1} + I \otimes \mathcal{Q}_1 \otimes I \otimes \cdots \otimes I + I \otimes \cdots \otimes I \otimes \mathcal{Q}_1 \,, \tag{45}$$

---

[4]This is the limit in which the theory is integrable and have little to do with black holes. The latter are obtained in a different limit where the charges scale with $N$ appropriately.

[5]A word $w_i$ of total length $L$ has norm $\mathcal{M}_{ii}$ that grows as $N^L$ to the leading order. One could thus define $\tilde{w}_i \equiv N^{-L/2} w_i$ and $\tilde{\mathcal{M}}_{ij} \equiv \langle \tilde{w}_i | \tilde{w}_j \rangle$ to make the strict large $N$ limit well-defined.

where $I$ is the identity matrix acting on the space of single-trace words. Using (42) and (44), we find

$$\langle w_i | Q | w_j \rangle = (\mathcal{M} \mathcal{Q})_{ij}. \tag{46}$$

Taking the Hermitian conjugate, we obtain

$$\langle w_i | Q^\dagger | w_j \rangle = (\mathcal{Q}^\dagger \mathcal{M})_{ij}. \tag{47}$$

Combining the above two formulae, we find

$$\langle w_i | H | w_j \rangle = 2 \left( \langle w_i | Q Q^\dagger | w_j \rangle + \langle w_i | Q^\dagger Q | w_j \rangle \right) = 2(\mathcal{M} \mathcal{Q} \mathcal{M}^{-1} \mathcal{Q}^\dagger \mathcal{M} + \mathcal{Q}^\dagger \mathcal{M} \mathcal{Q})_{ij}. \tag{48}$$

Now, we consider the action of the Hamiltonian $H$ on the BPS words $\vec{w}$,

$$H \vec{w} \equiv \vec{w} \mathcal{H}, \tag{49}$$

where the matrix $\mathcal{H}$ is given by

$$\mathcal{H} = 2(\mathcal{Q} \mathcal{M}^{-1} \mathcal{Q}^\dagger \mathcal{M} + \mathcal{M}^{-1} \mathcal{Q}^\dagger \mathcal{M} \mathcal{Q}). \tag{50}$$

In the large $N$ limit, using (43) and (45), we find that the matrix $\mathcal{H}$ is also block diagonal with the block $\mathcal{H}_n$ of the $n$-trace words given by the tensor product

$$\mathcal{H}_n = \mathcal{H}_1 \otimes \underbrace{I \otimes \cdots \otimes I}_{n-1} + I \otimes \mathcal{H}_1 \otimes I \otimes \cdots \otimes I + I \otimes \cdots \otimes I \otimes \mathcal{H}_1. \tag{51}$$

Hence, we can focus on the block $\mathcal{H}_1$, whose eigenvalues are the anomalous dimensions of single-trace operators, and (51) proves the statements in the first paragraph of this subsection.

# 4 $Q$ and $H$ as differential operators

We can represent the supercharge $Q$ and the one-loop Hamiltonian $H$ as (functional) differential operators in superfield space. Even though this is not the method used to obtain the anomalous dimensions in Section 5, this representation serves as a convenient tool for analytic computations.

From the action of supercharge $Q$ on the superfield $\Psi$ (18), we can represent $Q$ as a differential operator with respect to $\Psi$:

$$Q = \mathrm{Tr} \left( \Psi^2 \frac{\delta}{\delta \Psi} \right) \Big|_{z=0,\,\theta=0}, \tag{52}$$

where $\dfrac{\delta}{\delta \Psi}$ as an $N \times N$ matrix has components $\left( \dfrac{\delta}{\delta \Psi} \right)_{IJ} = \dfrac{\delta}{\delta \Psi_{JI}}$. More explicitly, $\dfrac{\delta}{\delta \Psi}$ is written in terms of the derivatives of the BPS letters as

$$\frac{\delta}{\delta \Psi} \equiv i \sum_{n=0}^{\infty} \frac{\overleftarrow{\partial^n}}{\partial z_{\dot{\alpha}_1} \cdots \partial z_{\dot{\alpha}_n}} \left[ \frac{\partial}{\partial (D_{\dot{\alpha}_1} \cdots D_{\dot{\alpha}_{n-1}} \lambda_{\dot{\alpha}_n})} + \frac{1}{2} \frac{\overleftarrow{\partial}}{\partial \theta_i} \frac{\partial}{\partial (D_{\dot{\alpha}_1} \cdots D_{\dot{\alpha}_n} \phi^i)} \right.$$
$$\left. - \frac{1}{4} \epsilon_{ijk} \frac{\overleftarrow{\partial^2}}{\partial \theta_i \partial \theta_j} \frac{\partial}{\partial (D_{\dot{\alpha}_1} \cdots D_{\dot{\alpha}_n} \psi_k)} - \frac{1}{4} \frac{\overleftarrow{\partial^3}}{\partial \theta_1 \partial \theta_2 \partial \theta_3} \frac{\partial}{\partial (D_{\dot{\alpha}_1} \cdots D_{\dot{\alpha}_n} f)} \right]. \tag{53}$$

Here, the rule for the derivative with a left arrow is, e.g.

$$\Psi^n \frac{\overleftarrow{\partial^3}}{\partial \theta_1 \partial \theta_2 \partial \theta_3} \Psi = \left[ \frac{\partial^3}{\partial \theta_1 \partial \theta_2 \partial \theta_3} (\Psi^n) \right] \Psi, \tag{54}$$

for any positive integer $n$. Similarly, $Q^\dagger$ at one-loop order has the differential representation

$$Q^\dagger = \frac{g_{YM}^2}{16\pi^2} \text{Tr}\left(\Psi \frac{\delta^2}{\delta\Psi^2}\right)\bigg|_{z=0,\,\theta=0}, \tag{55}$$

where the normalization of $Q^\dagger$ can be fixed by matching with the known one-loop anomalous dimension of the Konishi operator, as will be done momentarily.[6] Substituting the differential representations (52) and (55) into (7) gives the representation of the Hamiltonian $H$.

**Konishi multiplet** Consider the charge sector $(n_{z^+}, n_{z^-}, n_{\theta_1}, n_{\theta_2}, n_{\theta_3}) = (0, 0, 1, 1, 1)$, with the number of $\Psi$'s given by $Y = 3$. This Hilbert subspace is spanned by two operators,

$$\begin{aligned}
\text{Tr}(\partial_{\theta_1}\Psi\partial_{\theta_2}\Psi\partial_{\theta_3}\Psi) &\sim \text{Tr}(\phi^1\phi^2\phi^3), \\
\text{Tr}(\partial_{\theta_1}\Psi\partial_{\theta_3}\Psi\partial_{\theta_2}\Psi) &\sim \text{Tr}(\phi^1\phi^3\phi^2).
\end{aligned} \tag{56}$$

From $\frac{\partial}{\partial\theta_i}\Psi^2\big|_{z=0,\,\theta=0} = 0$, we immediately have

$$Q\,\text{Tr}(\phi^1\phi^2\phi^3) = Q\,\text{Tr}(\phi^1\phi^3\phi^2) = 0. \tag{57}$$

We can also compute the $Q^\dagger$ action by acting $\frac{\partial}{\partial\Psi}$ one after another, giving

$$\begin{aligned}
Q^\dagger\text{Tr}(\phi^1\phi^2\phi^3) = &-\frac{g_{YM}^2}{64\pi^2}\Bigg[\text{Tr}\left(\partial_1\partial_2\Psi\frac{\partial}{\partial\phi^2}\frac{\partial}{\partial\phi^1}\right)\text{Tr}\left(\phi^1\phi^2\phi^3\right) \\
&+\text{Tr}\left(\partial_2\partial_3\Psi\frac{\partial}{\partial\phi^3}\frac{\partial}{\partial\phi^2}\right)\text{Tr}\left(\phi^2\phi^3\phi^1\right) + \text{Tr}\left(\partial_3\partial_1\Psi\frac{\partial}{\partial\phi^1}\frac{\partial}{\partial\phi^3}\right)\text{Tr}\left(\phi^3\phi^1\phi^2\right)\Bigg]\Bigg|_{z=0,\,\theta=0} \\
=& -\frac{ig_{YM}^2}{32\pi^2}N\text{Tr}\left(\psi_i\phi^i\right).
\end{aligned} \tag{58}$$

Similarly,

$$Q^\dagger\text{Tr}(\phi^1\phi^3\phi^2) = \frac{ig_{YM}^2}{32\pi^2}N\text{Tr}(\psi_i\phi^i). \tag{59}$$

Now we can compute $H\,\text{Tr}(\phi^1\phi^2\phi^3) = 2QQ^\dagger\text{Tr}(\phi^1\phi^2\phi^3)$, where

$$\begin{aligned}
QQ^\dagger\text{Tr}(\phi^1\phi^2\phi^3) =& -\frac{ig_{YM}^2}{32\pi^2}N\text{Tr}(\Psi^2\frac{\partial}{\partial\Psi})\text{Tr}(\psi_i\phi^i)\bigg|_{z=0,\,\theta=0} \\
=& -\frac{ig_{YM}^2}{32\pi^2}N\left[-\frac{i}{2}\text{Tr}(\partial_2\partial_3\Psi^2\frac{\partial}{\partial\psi_1})\text{Tr}(\psi_i\phi^i)\right]\bigg|_{z=0,\,\theta=0} + \text{cyclic} \\
=& \frac{3g_{YM}^2N}{16\pi^2}[\text{Tr}(\phi^1\phi^2\phi^3) - \text{Tr}(\phi^1\phi^3\phi^2)],
\end{aligned} \tag{60}$$

and

$$QQ^\dagger\text{Tr}(\phi^1\phi^3\phi^2) = -\frac{3g_{YM}^2N}{16\pi^2}[\text{Tr}(\phi^1\phi^2\phi^3) - \text{Tr}(\phi^1\phi^3\phi^2)]. \tag{61}$$

The matrix representation of $H$ in this charge sector can be immediately read off to be

$$H = \frac{3g_{YM}^2N}{8\pi^2}\begin{pmatrix} 1 & -1 \\ -1 & 1 \end{pmatrix}. \tag{62}$$

---

[6]Note that the method in Section 3 does not require extra information to determine the overall normalization of $Q^\dagger$, $H$, and relatedly, the one-loop anomalous dimensions.

Table 1: The maximal $n$ of our computation, the number of operators, the number of $\mathcal{C}(\Delta)$ multiplets, and the number of distinct values of one-loop anomalous dimensions, for each $N$. The subscripts ST and MT stand for single- and multi-trace, respectively.

| $N$ | $n_{\max}$ | # BPS operators | # non-BPS operators | # $\mathcal{C}(\Delta)$ long multiplets | # distinct nonzero values |
|---|---|---|---|---|---|
| 2 | 24 | 17990 | 2202266 | 934 | 776 |
| 3 | 17 | 3808 | 185939 | 311 | 232 |
| 4 | 15 | 2051 | 56353 | 144 | 101 |
| $\infty_{\mathrm{ST}}$ | 22 | 1917 | 7359925 | 9264 | 4286 |
| $\infty_{\mathrm{MT}}$ | 22 | 1086343 | 20501627 | 27617 | 4385 |

The eigenvalues are

$$0, \quad \frac{3g_{\mathrm{YM}}^2 N}{4\pi^2}. \tag{63}$$

In fact, the eigenvector with a non-zero eigenvalue is proportional to $\mathrm{Tr}(\phi^1[\phi^2, \phi^3])$, which is a descendant of the Konishi operator $\mathrm{Tr}(\Phi^{mn}\Phi_{mn})$ [34], since

$$Q_-^4 Q_+^4 \mathrm{Tr}(\Phi^{mn}\Phi_{mn}) = 4iQ_-^4 \mathrm{Tr}(\phi^i\psi_i) = 24\mathrm{Tr}(\phi^1[\phi^2, \phi^3]). \tag{64}$$

One can easily compare (63) with the known one-loop anomalous dimension of the Konishi operator, e.g. in [35], to verify that the normalization of $Q^\dagger$ in (55) is correct.

## 5 Results and discussions

Using the machinery developed in Section 3, we systematically constructed and diagonalized the one-loop Hamiltonian in the classically-BPS sector, in increasing

$$n \equiv 2(3J_L + q_1 + q_2 + q_3), \tag{65}$$

for all charges up to the $n_{\max}$ indicated in Table 1. Also specified are the number of $\mathcal{C}(\Delta)$ long multiplets and the number of distinct nonzero values of one-loop anomalous dimensions. The data can be publicly accessed on https://github.com/yinhslin/bps-counting.

When analyzing the data, we can consider the spectrum of all operators, of $\mathcal{C}(\Delta)$ primaries, or of $\mathcal{C}(\Delta)$ multiplets.[7] To study the statistics of anomalous dimensions, it is natural to consider the spectrum of primaries or multiplets to avoid large degeneracies. We choose to consider the spectrum of primaries, since each primary has well-defined angular momenta and R-charges, whereas a multiplet contains many different charges. Holographically, $\mathcal{C}(\Delta)$ acts as boosts (and fermionic generalizations) on objects in the bulk (see e.g. [31] for a discussion), hence if we are interested in analyzing the "core" properties of objects (such as the near-horizon excitations of black holes), and not their motion in the ambient AdS, then it is certainly more natural to consider the spectrum of primaries.

---

[7]Recall that $\mathcal{C}(\Delta)$ primaries are defined as operators annihilated by all the superconformal supercharges $S$ in $\mathcal{C}(\Delta)$, so a single multiplet can contain multiple primaries.

## 5.1 Statistics of anomalous dimensions and hints of a gap

The results of the one-loop anomalous dimensions are presented as the smooth histograms defined by the density

$$\rho(\delta) \equiv \frac{1}{|\mathcal{I}|} \sum_{\delta_i \in \mathcal{I}} \frac{1}{\sqrt{2\pi}\sigma} \exp\left[-\frac{(\delta_i - \delta)^2}{2\sigma^2}\right], \tag{66}$$

where $\delta$ is the normalized one-loop anomalous dimension

$$\delta \equiv \frac{\pi^2 E}{g_{\text{YM}}^2 N}, \quad E : \text{Eigenvalue of } H = g_{\text{YM}}^2 D^{(2)}, \tag{67}$$

and $\mathcal{I}$ is the set of $\mathcal{C}(\Delta)$ long multiplets or the set of non-BPS $\mathcal{C}(\Delta)$ primary operators.

Figure 1 and 2 present smooth histograms of $\mathcal{C}(\Delta)$ long multiplets and non-BPS $\mathcal{C}(\Delta)$ primaries, for all charge sectors up to the maximal $n$ indicated in Table 1 for various gauge groups. To avoid interference among different charge sectors, Figure 3 presents smooth histograms of superconformal primary operators in select charge sectors. The spectrum of a fixed-charge sector does not depend on $n_{\text{max}}$, but that without fixing charges does.

The spectrum of one-loop anomalous dimensions supports our conjecture (15). The smooth histograms for the $\text{SU}(\infty)_{\text{MT}}$ spectrum exhibit no visible "gap" (see Section 2.3 for proper notion) above $\delta = 0$ consistent with (15) where the gap becomes zero when $N \to \infty$.[8] On the other hand, the smooth histograms for SU(2), SU(3), SU(4) exhibit clear "gaps" above $\delta = 0$. However, our current data is not enough to fit the value of $\widetilde{\Delta}^{(1)}$ in (15).

It is tempting to think that our conjectural existence of a gap is the highly stringy version of the gap in the spectrum of near-extremal black holes established in [20] by a gravitational path integral computation. In the non-BPS case, there is no definitive method to distinguish multi-gravitons from potential black holes, thus putting an asterisk on how much our data reflects stringy black hole physics.[9] This issue is often ignored because in the charge regime (13), black holes dominate the statistical ensemble anyway.

Finally, note that even though the energy $E$ and $n_{\text{max}}$ of our data set are quite large compared to $N^2$, as seen in Table 1, the individual angular momenta and R-charges are still smaller than $N^2$ once we distribute $n$ over the five charges $(J_1, J_2, q_1, q_2, q_3)$, as seen in e.g. Figure 3.

## 5.2 Smallest BPS black hole operator at weak coupling

In [9–11, 13–16], BPS operators were analyzed and constructed at the level of cohomology. The non-renormalization theorem of [14] dictates that exactly-BPS representatives of the cohomologies exist to all orders in perturbation theory. In this work, we computed the actual BPS operators at weak coupling by diagonalizing the Hamiltonian.

The most interesting cohomology found in [14] was for the smallest BPS black hole (non-multi-graviton) operator $O_{\text{bh}}$ with gauge group SU(2). It has charge $(n_{z^+}, n_{z^-}, n_{\theta_1}, n_{\theta_2}, n_{\theta_3}) = (0, 0, 4, 4, 4)$ and is septic in the fundamental fields. Let us present its actual weak coupling expression—not merely at the level of cohomology but as a concrete operator (which was also computed in simultaneous work [23]).

To all orders in perturbation theory, $O_{\text{bh}}$ takes the form

$$O_{\text{bh}} = O + Q O', \tag{68}$$

---

[8]We do not have a clear mathematical definition of "gap" for finite $N$ and charges, so the word is used in a qualitative sense in which visually the smooth histogram appears to exhibit a gap.

[9]However, see Appendix F of [17] for a proposed criterion, and some evidence that all classically-BPS non-BPS states in the same charge sector as a BPS black hole are *black-hole-like*.

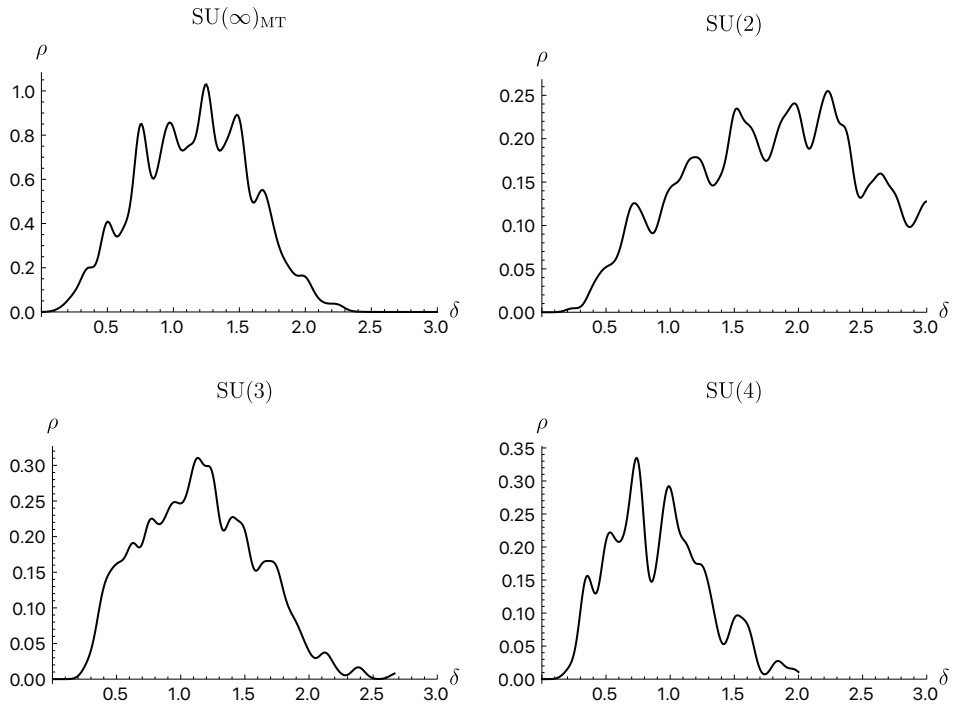

Figure 1: Smooth histograms (with $\sigma = 0.05$) of nonzero one-loop anomalous dimensions $\delta$ of $\mathcal{C}(\Delta)$ long multiplets, up to the maximal $n$ indicated in Table 1 for planar and $N = 2, 3, 4$.

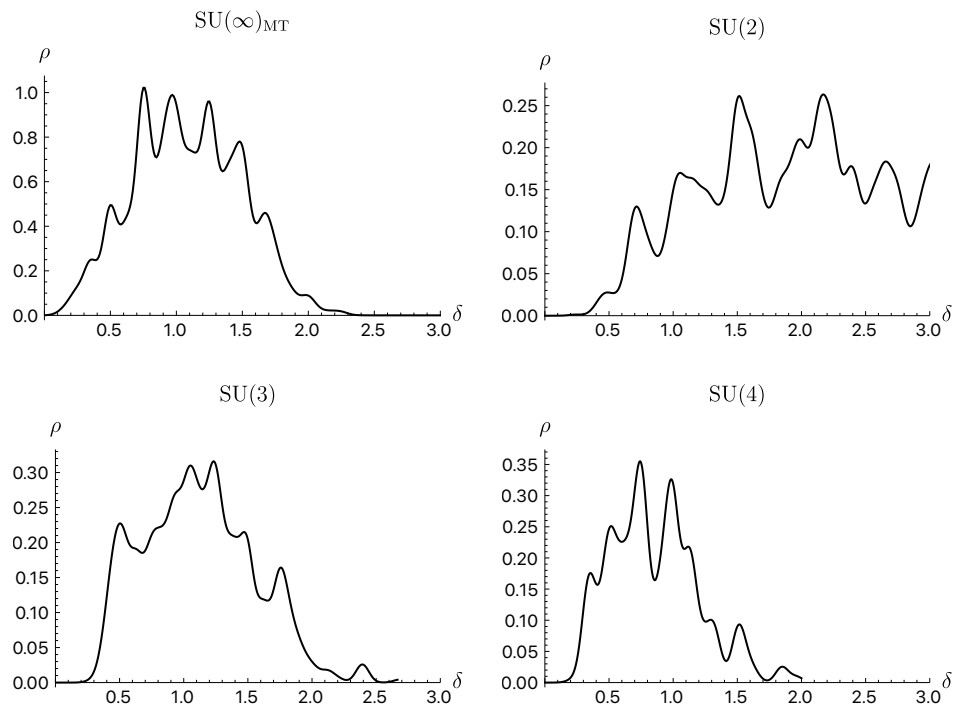

Figure 2: Smooth histograms (with $\sigma = 0.05$) of nonzero one-loop anomalous dimensions $\delta$ of non-BPS $\mathcal{C}(\Delta)$ primaries, up to the maximal $n$ indicated in Table 1 for planar and $N = 2, 3, 4$.



Figure 3: Smooth histograms (with $\sigma = 0.2$) of nonzero one-loop anomalous dimensions $\delta$ of the $\mathcal{C}(\Delta)$ primary operators with select charges labeled as $(J_L, J_R, q_1, q_2, q_3)$. The charge sectors on the left column ($SU(\infty)_{MT}$ cases) are at $n = 22$, and on the right column ($SU(2)$ cases) are at $n = 24$. The values of $\delta_{gap}$ displayed on top of the plots are the actual smallest values of $\delta$.

where $O$ is a representative of the 1-dimensional $Q$-cohomology (i.e. it is $Q$-closed but not $Q$-exact), and $O'$ has charge $(0, 0, 4, 4, 4)$ and is sextic in the fundamental fields. The space of (gauge-invariant) operators in the classically-BPS sector with charge $(0, 0, 4, 4, 4)$ and sextic in the fundamental fields is 53-dimensional, of which a 17-dimensional subspace is invariant under the cyclic symmetry permuting $\theta_i$. In [15], a basis for this 17-dimensional subspace was chosen to be[10]

$$
\begin{aligned}
O'_1 &= \mathrm{Tr}(\partial_{\theta_1}\partial_{\theta_2}\partial_{\theta_3}\Psi\partial_{\theta_1}\partial_{\theta_2}\partial_{\theta_3}\Psi)\mathrm{Tr}(\partial_{\theta_1}\partial_{\theta_2}\partial_{\theta_3}\Psi\partial_{\theta_1}\Psi)\mathrm{Tr}(\partial_{\theta_2}\Psi\partial_{\theta_2}\Psi) + \text{cyclic}, \\
O'_2 &= \mathrm{Tr}(\partial_{\theta_1}\partial_{\theta_2}\partial_{\theta_3}\Psi\partial_{\theta_1}\partial_{\theta_2}\partial_{\theta_3}\Psi)\mathrm{Tr}(\partial_{\theta_1}\Psi\partial_{\theta_2}\Psi)\mathrm{Tr}(\partial_{\theta_2}\partial_{\theta_3}\Psi\partial_{\theta_1}\partial_{\theta_3}\Psi) + \text{cyclic}, \\
O'_3 &= \mathrm{Tr}(\partial_{\theta_1}\partial_{\theta_2}\partial_{\theta_3}\Psi\partial_{\theta_1}\partial_{\theta_2}\partial_{\theta_3}\Psi)\mathrm{Tr}(\partial_{\theta_1}\Psi\partial_{\theta_2}\partial_{\theta_3}\Psi)\mathrm{Tr}(\partial_{\theta_2}\Psi\partial_{\theta_1}\partial_{\theta_3}\Psi) + \text{cyclic}, \\
O'_4 &= \mathrm{Tr}(\partial_{\theta_1}\partial_{\theta_2}\partial_{\theta_3}\Psi\partial_{\theta_1}\Psi)\mathrm{Tr}(\partial_{\theta_1}\partial_{\theta_2}\partial_{\theta_3}\Psi\partial_{\theta_2}\Psi)\mathrm{Tr}(\partial_{\theta_2}\partial_{\theta_3}\Psi\partial_{\theta_1}\partial_{\theta_3}\Psi) + \text{cyclic}, \\
O'_5 &= \mathrm{Tr}(\partial_{\theta_1}\partial_{\theta_2}\partial_{\theta_3}\Psi\partial_{\theta_2}\partial_{\theta_3}\Psi)\mathrm{Tr}(\partial_{\theta_1}\partial_{\theta_2}\partial_{\theta_3}\Psi\partial_{\theta_1}\partial_{\theta_3}\Psi)\mathrm{Tr}(\partial_{\theta_1}\Psi\partial_{\theta_2}\Psi) + \text{cyclic}, \\
O'_6 &= \mathrm{Tr}(\partial_{\theta_1}\partial_{\theta_2}\partial_{\theta_3}\Psi\partial_{\theta_1}\Psi)\mathrm{Tr}(\partial_{\theta_1}\partial_{\theta_2}\partial_{\theta_3}\Psi\partial_{\theta_2}\partial_{\theta_3}\Psi)\mathrm{Tr}(\partial_{\theta_2}\Psi\partial_{\theta_1}\partial_{\theta_3}\Psi) + \text{cyclic}, \\
O'_7 &= \mathrm{Tr}(\partial_{\theta_1}\partial_{\theta_2}\partial_{\theta_3}\Psi\partial_{\theta_2}\Psi)\mathrm{Tr}(\partial_{\theta_1}\partial_{\theta_2}\partial_{\theta_3}\Psi\partial_{\theta_2}\partial_{\theta_3}\Psi)\mathrm{Tr}(\partial_{\theta_1}\Psi\partial_{\theta_1}\partial_{\theta_3}\Psi) + \text{cyclic}, \\
O'_8 &= \mathrm{Tr}(\partial_{\theta_1}\partial_{\theta_2}\partial_{\theta_3}\Psi\partial_{\theta_1}\Psi)\mathrm{Tr}(\partial_{\theta_1}\partial_{\theta_2}\partial_{\theta_3}\Psi\partial_{\theta_1}\partial_{\theta_3}\Psi)\mathrm{Tr}(\partial_{\theta_2}\Psi\partial_{\theta_2}\partial_{\theta_3}\Psi) + \text{cyclic}, \\
O'_9 &= \mathrm{Tr}(\partial_{\theta_1}\partial_{\theta_2}\partial_{\theta_3}\Psi\partial_{\theta_2}\Psi)\mathrm{Tr}(\partial_{\theta_1}\partial_{\theta_2}\partial_{\theta_3}\Psi\partial_{\theta_1}\partial_{\theta_3}\Psi)\mathrm{Tr}(\partial_{\theta_1}\Psi\partial_{\theta_2}\partial_{\theta_3}\Psi) + \text{cyclic}, \\
O'_{10} &= \mathrm{Tr}(\partial_{\theta_1}\partial_{\theta_2}\partial_{\theta_3}\Psi\partial_{\theta_1}\partial_{\theta_2}\partial_{\theta_3}\Psi)\mathrm{Tr}(\partial_{\theta_1}\Psi\partial_{\theta_1}\partial_{\theta_3}\Psi)\mathrm{Tr}(\partial_{\theta_2}\Psi\partial_{\theta_2}\partial_{\theta_3}\Psi) + \text{cyclic}, \\
O'_{11} &= \mathrm{Tr}(\partial_{\theta_1}\partial_{\theta_2}\partial_{\theta_3}\Psi\partial_{\theta_1}\Psi)\mathrm{Tr}(\partial_{\theta_2}\partial_{\theta_3}\Psi\partial_{\theta_1}\partial_{\theta_3}\Psi)\mathrm{Tr}(\partial_{\theta_2}\partial_{\theta_3}\Psi\partial_{\theta_1}\partial_{\theta_2}\Psi) + \text{cyclic}, \\
O'_{12} &= \mathrm{Tr}(\partial_{\theta_1}\partial_{\theta_2}\partial_{\theta_3}\Psi\partial_{\theta_1}\partial_{\theta_3}\Psi)\mathrm{Tr}(\partial_{\theta_2}\partial_{\theta_3}\Psi\partial_{\theta_1}\Psi)\mathrm{Tr}(\partial_{\theta_2}\partial_{\theta_3}\Psi\partial_{\theta_1}\partial_{\theta_2}\Psi) + \text{cyclic}, \\
O'_{13} &= \mathrm{Tr}(\partial_{\theta_1}\partial_{\theta_2}\partial_{\theta_3}\Psi\partial_{\theta_1}\partial_{\theta_2}\Psi)\mathrm{Tr}(\partial_{\theta_2}\partial_{\theta_3}\Psi\partial_{\theta_1}\partial_{\theta_3}\Psi)\mathrm{Tr}(\partial_{\theta_2}\partial_{\theta_3}\Psi\partial_{\theta_1}\Psi) + \text{cyclic}, \\
O'_{14} &= \mathrm{Tr}(\partial_{\theta_1}\partial_{\theta_2}\partial_{\theta_3}\Psi\partial_{\theta_2}\partial_{\theta_3}\Psi)\mathrm{Tr}(\partial_{\theta_1}\Psi\partial_{\theta_1}\partial_{\theta_3}\Psi)\mathrm{Tr}(\partial_{\theta_2}\partial_{\theta_3}\Psi\partial_{\theta_1}\partial_{\theta_2}\Psi) + \text{cyclic}, \\
O'_{15} &= \mathrm{Tr}(\partial_{\theta_1}\partial_{\theta_2}\partial_{\theta_3}\Psi\partial_{\theta_1}\Psi)\mathrm{Tr}(\partial_{\theta_1}\partial_{\theta_2}\partial_{\theta_3}\Psi\partial_{\theta_2}\partial_{\theta_3}\Psi)\mathrm{Tr}(\partial_{\theta_1}\Psi\partial_{\theta_2}\partial_{\theta_3}\Psi) + \text{cyclic}, \\
O'_{16} &= \mathrm{Tr}(\partial_{\theta_1}\partial_{\theta_2}\partial_{\theta_3}\Psi\partial_{\theta_2}\partial_{\theta_3}\Psi)\mathrm{Tr}(\partial_{\theta_1}\partial_{\theta_3}\Psi\partial_{\theta_1}\partial_{\theta_2}\Psi)\mathrm{Tr}(\partial_{\theta_1}\Psi\partial_{\theta_2}\partial_{\theta_3}\Psi) + \text{cyclic}, \\
O'_{17} &= \mathrm{Tr}(\partial_{\theta_2}\partial_{\theta_3}\Psi\partial_{\theta_1}\partial_{\theta_3}\Psi)\mathrm{Tr}(\partial_{\theta_1}\partial_{\theta_3}\Psi\partial_{\theta_1}\partial_{\theta_2}\Psi)\mathrm{Tr}(\partial_{\theta_1}\partial_{\theta_2}\Psi\partial_{\theta_2}\partial_{\theta_3}\Psi),
\end{aligned}
\tag{70}
$$

and $O$ was chosen to be

$$
\begin{aligned}
O = \; & \mathrm{Tr}(\partial_{\theta_2}\partial_{\theta_3}\Psi\partial_{\theta_1}\Psi + \partial_{\theta_1}\partial_{\theta_3}\Psi\partial_{\theta_2}\Psi)\mathrm{Tr}(\partial_{\theta_1}\partial_{\theta_2}\Psi\partial_{\theta_1}\Psi)\mathrm{Tr}(\partial_{\theta_1}\partial_{\theta_3}\Psi\partial_{\theta_2}\partial_{\theta_3}\Psi\partial_{\theta_2}\partial_{\theta_3}\Psi) \\
& + \text{cyclic}.
\end{aligned}
\tag{71}
$$

By direct computation, we find that the actual BPS black hole operator at one-loop order has the expression

$$
\begin{aligned}
O_{\mathrm{bh}} = \; & O + \frac{3O_3}{40} + \frac{O_6}{180} - \frac{O_7}{20} - \frac{O_8}{20} + \frac{O_9}{180} + \frac{3O_{10}}{40} \\
& + \frac{O_{12}}{9} + \frac{17O_{13}}{90} + \frac{3O_{14}}{10} + \frac{2O_{15}}{45} - \frac{O_{16}}{9} + \frac{O_{17}}{2},
\end{aligned}
\tag{72}
$$

where $O_i \equiv QO'_i$ for $i = 1, \ldots, 17$.

Let us write the BPS black hole operator $O_{\mathrm{bh}}$ in a manifestly $SU(3)$ invariant form. Let us consider the $SU(3)$ invariant representative in [16],

$$
\begin{aligned}
\widetilde{O} = \; & \epsilon_{i_1 i_2 i_3}\epsilon_{j_1 j_2 j_3}\epsilon_{k_1 k_2 k_3}\epsilon_{l_1 l_2 l_3}\epsilon_{m_1 m_2 m_3}\epsilon^{k_1 l_1 m_1} \\
& \times \mathrm{Tr}(\partial^{i_1}\Psi\partial^{k_2 k_3}\Psi)\mathrm{Tr}(\partial^{j_1}\Psi\partial^{l_2 l_3}\Psi)\mathrm{Tr}(\partial^{i_2 i_3}\Psi\partial^{j_2 j_3}\Psi\partial^{m_2 m_3}\Psi),
\end{aligned}
\tag{73}
$$

---

[10]In the early versions of [15], $O'_{15}$ was written as (up to normalization)

$$
\mathrm{Tr}(\partial_{\theta_1}\partial_{\theta_2}\partial_{\theta_3}\Psi\partial_{\theta_2}\partial_{\theta_3}\Psi)\mathrm{Tr}(\partial_{\theta_1}\Psi\partial_{\theta_1}\partial_{\theta_2}\Psi)\mathrm{Tr}(\partial_{\theta_2}\partial_{\theta_3}\Psi\partial_{\theta_1}\partial_{\theta_3}\Psi) + \text{cyclic},
\tag{69}
$$

which equals $-\frac{1}{2}O'_3 - \frac{1}{2}O'_{10} - O'_{12} - O'_{13} - O'_{14}$. To fix this, we replaced (69) with the expression in (70).

where we used the abbreviation $\partial^{i_1 \cdots i_n} \equiv \partial_{\theta_{i_1}} \cdots \partial_{\theta_{i_n}}$. The SU(3) invariant subspace is four-dimensional and we choose the basis to be

$$
\begin{aligned}
\widetilde{O}'_1 &= \epsilon_{ijk}\epsilon_{lmn}\text{Tr}(\partial^{123}\Psi\partial^i\Psi)\text{Tr}(\partial^{123}\Psi\partial^{jk}\Psi)\text{Tr}(\partial^l\Psi\partial^{mn}\Psi), \\
\widetilde{O}'_2 &= \epsilon_{imn}\epsilon_{jkl}\text{Tr}(\partial^{123}\Psi\partial^i\Psi)\text{Tr}(\partial^{123}\Psi\partial^{jk}\Psi)\text{Tr}(\partial^l\Psi\partial^{mn}\Psi), \\
\widetilde{O}'_3 &= \epsilon_{i_1i_2i_3}\epsilon^{j_1k_1l_1}\epsilon_{j_2j_3}\epsilon_{k_1k_2k_3}\epsilon_{l_1l_2l_3}\text{Tr}(\partial^{123}\Psi\partial^{i_1i_2}\Psi)\text{Tr}(\partial^{i_3}\Psi\partial^{j_2j_3}\Psi)\text{Tr}(\partial^{k_2k_3}\Psi\partial^{l_2l_3}\Psi), \\
\widetilde{O}'_4 &= \epsilon_{i_1i_2i_3}\epsilon^{j_1k_1l_1}\epsilon_{j_2j_3}\epsilon_{k_1k_2k_3}\epsilon_{l_1l_2l_3}\text{Tr}(\partial^{123}\Psi\partial^{j_2j_3}\Psi)\text{Tr}(\partial^{i_1}\Psi\partial^{k_2k_3}\Psi)\text{Tr}(\partial^{i_2i_3}\Psi\partial^{l_2l_3}\Psi).
\end{aligned}
\tag{74}
$$

The BPS black hole operator $O_{\text{bh}}$ can be written as

$$
O_{\text{bh}} = \widetilde{O} - \frac{4\widetilde{O}_1}{9} + 4\widetilde{O}_2 - \frac{7\widetilde{O}_3}{9} + \frac{10\widetilde{O}_4}{9}, \tag{75}
$$

where $\widetilde{O}_i \equiv Q\widetilde{O}'_i$ for $i = 1, \cdots, 4$.

# 6 Outlook

In this work, the superspace formulation of [13, 14] describing the classically-supersymmetric sector of $\mathcal{N} = 4$ super-Yang-Mills has been extended to capture non-supersymmetric aspects of the theory. The formalism's striking simplicity promises new progress in the old perturbative approach to interactions. Concretely, we used the formalism to amass a large data set of one-loop anomalous dimensions capturing near-supersymmetric black holes in a highly stringy regime.

It would be highly desirable if the full symmetry algebra $\mathcal{C}(\Delta)$ could be manifest in the Hamiltonian. Our current approach requires first constructing and diagonalizing the Hamiltonian involving all operators, and then *a posteriori* organizing the results into $\mathcal{C}(\Delta)$ multiplets. As is clear from Table 1, the actual number of multiplets or primaries is much smaller than that of all operators, so there appears to be a large amount of redundant effort. Eliminating this redundancy could be key in pushing the effectiveness of the superspace formalism.[11]

The one-loop Hamiltonian can also be viewed as a non-relativistic reduction of $\mathcal{N} = 4$ super-Yang-Mills. Treating this Hamiltonian as a quantum mechanical theory in its own right is a subject dubbed Spin-Matrix theory [36]. The superspace formalism may provide a useful restructuring of the quantum mechanics.

Another natural question to ask is whether the superspace formalism applies to higher loops. The requirement that the superconformal supercharges $S$ and special conformal generators $K$ should admit series expansions in $g_{\text{YM}}$ compatible with the $\mathcal{C}(\Delta)$ superconformal algebra and the non-renormalization of the supercharges $Q$ and momenta $P$ might be enough to "bootstrap" $g_{\text{YM}}$ corrections to the differential representation of $Q^\dagger$ at higher loops. In smaller subsectors, compatibility with the $\mathfrak{psu}(2,2|4)$ superconformal algebra has been used to determine the $g_{\text{YM}}$ expansion of the dilatation operator $D$ to two and three-loop orders [24, 37–39].[12]

Jackiw-Teitelboim (JT) gravity is holographically dual to a random matrix theory with Gaussian unitary ensemble (GUE) [40]. This duality is further generalized to $\mathcal{N} = 1$ and $\mathcal{N} = 2$ JT supergravities [41, 42]. Can the random matrix dual of $\mathcal{N} = 2$ JT supergravity be embedded in $\mathcal{N} = 4$ SYM? At large $N$, consider a finite range of charge sectors with large angular momenta and R-charges

$$
J_1, J_2, q_1, q_2, q_3 \sim N^2, \quad \Delta J_1, \Delta J_2, \Delta q_1, \Delta q_2, \Delta q_3 \sim 1. \tag{76}
$$

---

[11]First constructing all operators and then selecting primaries by imposing the $S = 0$ conditions is not too helpful, as the first step would impose a computational bottleneck.

[12]The supercharge $Q$ receives superficial $g_{\text{YM}}$ corrections (see (4.1) in [38]) that can be eliminated by a change of basis.

Perhaps the Hamiltonians over these charge sectors can be regarded as random matrices with varying dimensionality. Can this offer a connection to the random matrix dual of JT?

We leave the reader with a few more open questions. Is there a gauge-theoretic way to distinguish graviton operators from black holes in the near-BPS sector? Can the weak gauge coupling regime be accessed by first-principle gravitational techniques? For instance, does the proposed dual of free Yang-Mills theory in [43] have modifications/deformations that could activate $g_{YM}$?

## Acknowledgments

We thank Sunjin Choi, Luca Iliesiu, Jun Nian, Cheng Peng, and Elli Pomoni for helpful discussions, as well as Kasia Budzik, Harish Murali, and Pedro Vieira for coordinating the submission to the arXiv of related work [23].

**Funding information** CC and YT are partly supported by National Key R&D Program of China (NO. 2020YFA0713000). LF is supported in part by the NSFC under grant No. 12147103. YL is supported by the Simons Collaboration Grant on the Non-Perturbative Bootstrap. CC thanks Korea Institute for Advanced Study and the "Entanglement, Large N and Black Hole" workshop hosted by Asia Pacific Center for Theoretical Physics, LF thanks the "Forum on Black Holes and Quantum Entanglement" hosted by International Centre for Theoretical Physics Asia-Pacific, and YL thanks the "Matrix Models and String Field Theory" hosted by Centro de Ciencias de Benasque Pedro Pascual, as well as the "50 Years of Supersymmetry (SUSY50)" workshop hosted by Fine Theoretical Physics Institute, for hospitality during the progression of this work. The computations in this paper were run on the FASRC Cannon cluster supported by the FAS Division of Science Research Computing Group at Harvard University.

## A  Spinor convention

We define

$$\sigma^0 = \begin{pmatrix} 1 & 0 \\ 0 & 1 \end{pmatrix}, \quad \sigma^1 = i\begin{pmatrix} 0 & 1 \\ 1 & 0 \end{pmatrix}, \quad \sigma^2 = i\begin{pmatrix} 0 & -i \\ i & 0 \end{pmatrix}, \quad \sigma^3 = i\begin{pmatrix} 1 & 0 \\ 0 & -1 \end{pmatrix}. \tag{A.1}$$

The components satisfy

$$(\sigma^\mu)^*_{\alpha\dot\alpha} = \epsilon^{\alpha\beta}\epsilon^{\dot\alpha\dot\beta}(\sigma^\mu)_{\beta\dot\beta}. \tag{A.2}$$

We have the identities

$$(\sigma^\mu)_{\alpha\dot\alpha}(\sigma_\mu)_{\beta\dot\beta} = 2\epsilon_{\alpha\beta}\epsilon_{\dot\alpha\dot\beta}, \quad (\sigma^\mu)_{\alpha\dot\alpha}(\sigma_\mu)^{\beta\dot\beta} = 2\delta^\beta_\alpha\delta^{\dot\beta}_{\dot\alpha}, \quad (\sigma^\mu)^{\alpha\dot\alpha}(\sigma^\nu)_{\alpha\dot\alpha} = 2\delta^{\mu\nu}. \tag{A.3}$$

We use the convention $\epsilon^{+-} = 1 = \epsilon_{+-}$, and

$$v^\alpha = \epsilon^{\alpha\beta}v_\beta, \quad v_\alpha = v^\beta\epsilon_{\beta\alpha}, \quad v^{\dot\alpha} = \epsilon^{\dot\alpha\dot\beta}v_{\dot\beta}, \quad v_{\dot\alpha} = v^{\dot\beta}\epsilon_{\dot\beta\dot\alpha}. \tag{A.4}$$

Let us define

$$(\sigma_{\mu\nu})^\alpha_\beta = \frac{1}{2}(\sigma_{[\mu})^{\alpha\dot\gamma}(\sigma_{\nu]})_{\beta\dot\gamma}, \quad (\sigma_{\mu\nu})^{\dot\alpha}_{\dot\beta} = \frac{1}{2}(\sigma_{[\mu})^{\gamma\dot\alpha}(\sigma_{\nu]})_{\gamma\dot\beta}, \tag{A.5}$$

which satisfies the SO(4) Lie algebra.

# B  Superconformal algebra

The $\mathfrak{psu}(2,2|4)$ superconformal algebra has a bosonic subalgebra is $\mathfrak{su}(2,2) \times \mathfrak{su}(4)$. The conformal algebra $\mathfrak{su}(2,2)$ is generated by the

$$P_{\alpha\dot\beta}, \quad K^{\alpha\dot\beta}, \quad D, \quad (J_L)^\alpha_\beta, \quad (J_R)^{\dot\alpha}_{\dot\beta}, \tag{B.1}$$

where $\alpha, \beta = +, -$ and $\dot\alpha, \dot\beta = \dot+, \dot-$ are the spinor indices of $\mathfrak{su}(2)_L$ and $\mathfrak{su}(2)_R$. The $\mathfrak{su}(4)$ R-symmetry is generated by

$$R^m_n. \tag{B.2}$$

with the traceless condition

$$R^1_1 + R^2_2 + R^3_3 + R^4_4 = 0. \tag{B.3}$$

The upper (lower) $m, n = 1, \cdots, 4$ are the (anti-)fundamental indices of $\mathfrak{su}(4)$. The fermionic generators of $\mathfrak{psu}(2,2|4)$ are

$$Q^n_\alpha, \quad S^\alpha_n, \quad \overline{Q}_{\dot\alpha n}, \quad \overline{S}^{\dot\alpha n}. \tag{B.4}$$

The nonzero commutators of the $\mathfrak{psu}(2,2|4)$ superconformal algebra are

$$[(J_L)^\alpha_\beta, (J_L)^\gamma_\delta] = \delta^\gamma_\beta (J_L)^\alpha_\delta - \delta^\alpha_\delta (J_L)^\gamma_\beta, \qquad [(J_R)^{\dot\alpha}_{\dot\beta}, (J_R)^{\dot\gamma}_{\dot\delta}] = \delta^{\dot\gamma}_{\dot\beta}(J_R)^{\dot\alpha}_{\dot\delta} - \delta^{\dot\alpha}_{\dot\delta}(J_R)^{\dot\gamma}_{\dot\beta},$$

$$[(J_L)^\alpha_\beta, P_{\gamma\dot\delta}] = -\delta^\alpha_\gamma P_{\beta\dot\delta} + \frac{1}{2}\delta^\alpha_\beta P_{\gamma\dot\delta}, \qquad [(J_L)^\alpha_\beta, K^{\gamma\dot\delta}] = \delta^\gamma_\beta K^{\alpha\dot\delta} - \frac{1}{2}\delta^\alpha_\beta K^{\gamma\dot\delta},$$

$$[(J_R)^{\dot\alpha}_{\dot\beta}, P_{\gamma\dot\delta}] = -\delta^{\dot\alpha}_{\dot\delta}P_{\gamma\dot\beta} + \frac{1}{2}\delta^{\dot\alpha}_{\dot\beta}P_{\gamma\dot\delta}, \qquad [(J_R)^{\dot\alpha}_{\dot\beta}, K^{\gamma\dot\delta}] = \delta^{\dot\delta}_{\dot\beta}K^{\gamma\dot\alpha} - \frac{1}{2}\delta^{\dot\alpha}_{\dot\beta}K^{\gamma\dot\delta},$$

$$[D, P_{\alpha\dot\beta}] = P_{\alpha\dot\beta}, \qquad [D, K^{\alpha\dot\beta}] = -K^{\alpha\dot\beta},$$

$$[K^{\alpha\dot\beta}, P_{\gamma\dot\delta}] = \delta^\gamma_\alpha \delta^{\dot\beta}_{\dot\delta} D - \delta^{\dot\beta}_{\dot\delta}(J_L)^\alpha_\gamma - \delta^\gamma_\alpha (J_R)^{\dot\beta}_{\dot\delta},$$

$$[K^{\alpha\dot\beta}, Q^n_\gamma] = \delta^\alpha_\gamma \overline{S}^{\dot\beta n}, \qquad [K^{\alpha\dot\beta}, \overline{Q}_{\dot\gamma n}] = \delta^{\dot\beta}_{\dot\gamma} S^\alpha_n,$$

$$[P_{\alpha\dot\beta}, S^\gamma_n] = -\delta^\gamma_\alpha \overline{Q}_{\dot\beta n}, \qquad [P_{\alpha\dot\beta}, \overline{S}^{\dot\gamma n}] = -\delta^{\dot\gamma}_{\dot\beta} Q^n_\alpha,$$

$$[(J_L)^\alpha_\beta, Q^n_\gamma] = -\delta^\alpha_\gamma Q^n_\beta + \frac{1}{2}\delta^\alpha_\beta Q^n_\gamma, \qquad [(J_L)^\alpha_\beta, S^\gamma_n] = \delta^\gamma_\beta S^\alpha_n - \frac{1}{2}\delta^\alpha_\beta S^\gamma_n,$$

$$[(J_R)^{\dot\alpha}_{\dot\beta}, \overline{Q}_{\dot\gamma n}] = -\delta^{\dot\alpha}_{\dot\gamma}\overline{Q}_{\dot\beta n} + \frac{1}{2}\delta^{\dot\alpha}_{\dot\beta}\overline{Q}_{\dot\gamma n}, \qquad [(J_R)^{\dot\alpha}_{\dot\beta}, \overline{S}^{\dot\gamma n}] = \delta^{\dot\gamma}_{\dot\beta}\overline{S}^{\dot\alpha n} - \frac{1}{2}\delta^{\dot\alpha}_{\dot\beta}\overline{S}^{\dot\gamma n}, \tag{B.5}$$

$$\{S^\alpha_m, Q^n_\beta\} = \frac{1}{2}\delta^n_m \delta^\alpha_\beta D - \delta^n_m (J_L)^\alpha_\beta - \delta^\alpha_\beta R^n_m,$$

$$\{\overline{S}^{\dot\alpha m}, \overline{Q}_{\dot\beta n}\} = \frac{1}{2}\delta^m_n \delta^{\dot\alpha}_{\dot\beta} D - \delta^m_n (J_R)^{\dot\alpha}_{\dot\beta} + \delta^{\dot\alpha}_{\dot\beta} R^m_n,$$

$$\{Q^m_\alpha, \overline{Q}_{\dot\beta n}\} = \delta^m_n P_{\alpha\dot\beta}, \qquad \{S^\alpha_m, \overline{S}^{\dot\beta n}\} = \delta^n_m K^{\alpha\dot\beta},$$

$$[R^m_n, R^p_q] = \delta^p_n R^m_q - \delta^m_q R^p_n,$$

$$[R^m_n, Q^p_\gamma] = \delta^p_n Q^m_\gamma - \frac{1}{4}\delta^m_n Q^p_\gamma, \qquad [R^m_n, \overline{Q}_{\dot\gamma p}] = -\delta^m_p \overline{Q}_{\dot\gamma n} + \frac{1}{4}\delta^m_n \overline{Q}_{\dot\gamma p},$$

$$[R^m_n, S^\gamma_p] = -\delta^m_p S^\gamma_n + \frac{1}{4}\delta^m_n S^\gamma_p, \qquad [R^m_n, \overline{S}^{\dot\gamma p}] = \delta^p_n \overline{S}^{\dot\gamma m} - \frac{1}{4}\delta^m_n \overline{S}^{\dot\gamma p},$$

$$[D, Q^m_\alpha] = \frac{1}{2}Q^m_\alpha, \qquad [D, \overline{Q}_{\dot\alpha m}] = \frac{1}{2}\overline{Q}_{\dot\alpha m},$$

$$[D, S^\alpha_m] = -\frac{1}{2}S^\alpha_m, \qquad [D, \overline{S}^{\dot\alpha m}] = -\frac{1}{2}\overline{S}^{\dot\alpha m}.$$

Let us denote the Hermitian conjugate in the radial quantization (BPZ conjugate) by †, and the Hermitian conjugate in the usual quantization (with the time translation generated by $P_0$) by $*$. We have

$$P^\dagger_{\alpha\dot\beta} = K^{\alpha\dot\beta}, \quad (Q^m_\alpha)^\dagger = S^\alpha_m, \quad (\overline{Q}_{\dot\alpha m})^\dagger = \overline{S}^{\dot\alpha m}, \tag{B.6}$$

and the other generators are †-conjugated to themselves. We also have

$$(Q^m_\alpha)^* = \overline{Q}_{\dot\alpha m}, \quad (S^\alpha_m)^* = \overline{S}^{\dot\alpha m}, \quad P^*_{\alpha\dot\beta} = P_{\beta\dot\alpha}, \quad (K^{\alpha\dot\beta})^* = K^{\beta\dot\alpha}, \quad ((J_L)^\alpha_\beta)^* = ((J_R)^{\dot\alpha}_{\dot\beta})^*. \tag{B.7}$$

and the other generators are $*$-conjugated to themselves.

It is sometimes convenient to use a different parametrization of the Cartan generators as

$$q_1 = -R^2_2 - R^3_3, \qquad q_2 = -R^1_1 - R^3_3, \qquad q_3 = -R^1_1 - R^2_2, \tag{B.8}$$

and

$$(J_L)^-_- \equiv J_L \equiv \frac{J_1 + J_2}{2}, \qquad (J_R)^{\dot{}}_{\dot{}} \equiv J_R \equiv \frac{J_1 - J_2}{2}. \tag{B.9}$$

$q_i$ and $J_i$ generate rotations along the five orthogonal two-planes inside $\mathbb{R}^{10}$ where the $\mathfrak{so}(6) \times \mathfrak{so}(4) \cong \mathfrak{su}(4) \times \mathfrak{su}(2)_L \times \mathfrak{su}(2)_R$ act. The supercharges $Q^m_\alpha$ and $\overline{Q}_{\dot\alpha m}$ can be relabeled using the eigenvalues of $q_i$ and $J_i$ as $Q^{q_1,q_2,q_3}_{J_1,J_2}$,

$$\begin{aligned}
Q^1_\pm = Q^{+,-,-}_{\pm,\pm}, \qquad Q^2_\pm = Q^{-,+,-}_{\pm,\pm}, \qquad Q^3_\pm = Q^{-,-,+}_{\pm,\pm}, \qquad Q^4_\pm = Q^{+,+,+}_{\pm,\pm}, \\
\overline{Q}_{\pm 1} = Q^{-,+,+}_{\pm,\mp}, \qquad \overline{Q}_{\pm 2} = Q^{+,-,+}_{\pm,\mp}, \qquad \overline{Q}_{\pm 3} = Q^{+,+,-}_{\pm,\mp}, \qquad \overline{Q}_{\pm 4} = Q^{-,-,-}_{\pm,\mp}.
\end{aligned} \tag{B.10}$$

Another commonly used parametrization of the Cartan generators of $\mathfrak{su}(4)$ is the Dynkin labels $R_1$, $R_2$, and $R_3$. They are related to $q_i$ and $R^m_m$ by

$$R_1 = R^1_1 - R^2_2 = q_1 - q_2, \quad R_2 = R^2_2 - R^3_3 = q_2 - q_3, \quad R_3 = R^3_3 - R^4_4 = -q_1 - q_2. \tag{B.11}$$

Finally, the momentum, special conformal, and Lorentz generators with the vector indices ($P_\mu$, $K_\mu$, and $M_{\mu\nu}$) are given by

$$\begin{aligned}
P_{\alpha\dot\beta} = \frac{1}{2}(\sigma^\mu)_{\alpha\dot\beta} P_\mu, \qquad & K^{\alpha\dot\beta} = \frac{1}{2}(\sigma^\mu)^{\alpha\dot\beta} K_\mu, \\
(J_L)^\alpha_\beta = \frac{1}{2}(\sigma^{\mu\nu})^\alpha_\beta M_{\mu\nu}, \qquad & (J_R)^{\dot\alpha}_{\dot\beta} = \frac{1}{2}(\sigma_{\mu\nu})^{\dot\alpha}_{\dot\beta} M_{\mu\nu}.
\end{aligned} \tag{B.12}$$

They satisfy the standard commutation relations

$$\begin{aligned}
[M_{\mu\nu}, M_{\rho\sigma}] &= \delta_{\nu\rho} M_{\mu\sigma} - \delta_{\mu\rho} M_{\nu\sigma} + \delta_{\nu\sigma} M_{\rho\mu} - \delta_{\mu\sigma} M_{\rho\nu}, \\
[M_{\mu\nu}, P_\rho] &= \delta_{\nu\rho} P_\mu - \delta_{\mu\rho} P_\nu, \qquad [M_{\mu\nu}, K_\rho] = \delta_{\nu\rho} K_\mu - \delta_{\mu\rho} K_\nu, \\
[K_\mu, P_\nu] &= 2\delta_{\mu\nu} D - 2M_{\mu\nu}.
\end{aligned} \tag{B.13}$$

## C Centralizer subalgebra $\mathcal{C}(\Delta)$

We list the generators of the centralizer $\mathcal{C}(\Delta) = \mathfrak{u}(1) \ltimes [\mathfrak{psu}(1,2|3) \times \mathfrak{su}(1|1)]$. The $\mathfrak{u}(1)$ is generated by

$$(J_L)^-_-. \tag{C.1}$$

The $\mathfrak{su}(1|1)$ is generated by

$$Q^4_-, \quad S^-_4, \quad 2\{Q^4_-, S^-_4\} = D - 2(J_L)^-_- - 2R^4_4. \tag{C.2}$$

The $\mathfrak{psu}(1,2|3)$ has bosonic subalgebra $\mathfrak{su}(1,2) \times \mathfrak{su}(3)$. The $\mathfrak{su}(3)$ is generated by

$$\mathcal{R}^i_j \equiv R^i_j - \frac{1}{3}\delta^i_j(R^1_1 + R^2_2 + R^3_3). \tag{C.3}$$

The $\mathfrak{su}(1,2)$ is generated by

$$(J_R)^{\dot\alpha}_{\dot\beta}, \quad P_{\dot\alpha} \equiv P_{+\dot\alpha}, \quad K^{\dot\alpha} \equiv K^{+\dot\alpha}, \quad D + (J_L)^-_-. \tag{C.4}$$

The fermionic generators of $\mathfrak{psu}(1,2|3)$ are

$$Q^i_+, \quad \overline{Q}_{\dot\alpha i}, \quad S^+_i, \quad \overline{S}^{\dot\alpha i}. \tag{C.5}$$

The centralizer $\mathcal{C}(\Delta)$ has a $u(1,2)$ subalgebra, whose commutators are

$$M^{\dot\alpha}{}_{\dot\beta} \equiv \frac{1}{2}(\sigma^\mu)^{+\dot\alpha}(\sigma^\nu)_{+\dot\beta}M_{\mu\nu} = (J_R)^{\dot\alpha}_{\dot\beta} - (J_L)^-_-\delta^{\dot\alpha}_{\dot\beta}, \tag{C.6}$$

where $M_{\mu\nu}$ is the generators of the conformal algebra $so(2,4)$. We have the commutators

$$
\begin{aligned}
&[K^{\dot\alpha}, P_{\dot\beta}] = \delta^{\dot\alpha}_{\dot\beta}D - M^{\dot\alpha}{}_{\dot\beta}, &\quad& [M^{\dot\alpha}{}_{\dot\beta}, M^{\dot\gamma}{}_{\dot\delta}] = \delta^{\dot\gamma}_{\dot\beta}M^{\dot\alpha}{}_{\dot\delta} - \delta^{\dot\alpha}_{\dot\delta}M^{\dot\gamma}{}_{\dot\beta}, \\
&[M^{\dot\alpha}{}_{\dot\beta}, P_{\dot\gamma}] = -\delta^{\dot\alpha}_{\dot\gamma}P_{\dot\beta}, &\quad& [M^{\dot\alpha}{}_{\dot\beta}, K^{\dot\gamma}] = \delta^{\dot\gamma}_{\dot\beta}K^{\dot\alpha}.
\end{aligned}
\tag{C.7}
$$

# D  Fundamental fields in $\mathcal{N}=4$ SYM

The fundamental fields in $\mathcal{N}=4$ SYM are

$$\Phi_{mn}, \quad \Psi_{\alpha m}, \quad \overline{\Psi}^m_{\dot\alpha}, \quad A_{\alpha\dot\beta} \equiv \frac{1}{4}(\sigma^\mu)_{\alpha\dot\beta}A_\mu. \tag{D.1}$$

The scalars $\Phi_{mn}$ satisfy the reality condition $\Phi^*_{mn} = \frac{1}{2}\epsilon^{mnpq}\Phi_{pq} \equiv \Phi^{mn}$, and the fermions $\Psi_{\alpha m}$ and $\overline{\Psi}^m_{\dot\alpha}$ are $*$-conjugates of each other, i.e. $\Psi^*_{\alpha m} = \overline{\Psi}^m_{\dot\alpha}$.

The supercharges $Q^m_\alpha$ and $\overline{Q}_{\dot\alpha m}$ act on the fundamental fields as [11, 24, 44][13]

$$
\begin{aligned}
&[Q^m_\alpha, \Phi_{np}] = 2\delta^m_{[n}\Psi_{p]\alpha}, &\quad& \{Q^m_\alpha, \overline{\Psi}^n_{\dot\beta}\} = 2iD_{\alpha\dot\beta}\Phi^{mn}, \\
&\{Q^m_\alpha, \Psi_{\beta n}\} = -2i\delta^m_n F_{\alpha\beta} + \epsilon_{\alpha\beta}[\Phi^{mp}, \Phi_{np}], &\quad& [Q^m_\alpha, A_{\beta\dot\gamma}] = -\epsilon_{\alpha\beta}\overline{\Psi}^m_{\dot\gamma}, \\
&[\overline{Q}_{m\dot\alpha}, \Phi^{np}] = -2\delta^{[n}_m\overline{\Psi}^{p]}_{\dot\alpha}, &\quad& \{\overline{Q}_{\dot\alpha m}, \Psi_{\beta n}\} = 2iD_{\beta\dot\alpha}\Phi_{mn}, \\
&\{\overline{Q}_{\dot\alpha m}, \overline{\Psi}^n_{\dot\beta}\} = -2i\delta^n_m F_{\dot\alpha\dot\beta} - \epsilon_{\dot\alpha\dot\beta}[\Phi^{np}, \Phi_{mp}], &\quad& [\overline{Q}_{\dot\alpha m}, A_{\beta\dot\gamma}] = -\epsilon_{\dot\alpha\dot\gamma}\Psi_{\beta m},
\end{aligned}
\tag{D.3}
$$

where

$$
\begin{aligned}
&F_{\alpha\beta} \equiv -\frac{1}{16}(\sigma^{\mu\nu})_{\alpha\beta}F_{\mu\nu}, &\quad& F_{\dot\alpha\dot\beta} \equiv -\frac{1}{16}(\sigma^{\mu\nu})_{\dot\alpha\dot\beta}F_{\mu\nu}, \\
&\partial_{\alpha\dot\beta} = \frac{1}{4}(\sigma^\mu)_{\alpha\dot\beta}\partial_\mu, &\quad& D_{\alpha\dot\beta} = \frac{1}{4}(\sigma^\mu)_{\alpha\dot\beta}D_\mu, \quad D_\mu = \partial_\mu - iA_\mu.
\end{aligned}
\tag{D.4}
$$

---

[13]Our convention is related to the convention in [24] by

$$
\begin{aligned}
&\overline{\Psi}^{\text{them},m}_{\dot\alpha} = \frac{i}{2g}\overline{\Psi}^{\text{us},m}_{\dot\alpha}, &\quad& \Psi^{\text{them}}_{\alpha m} = \frac{i}{2g}\Psi^{\text{us}}_{\alpha m}, &\quad& \Phi^{\text{them}}_m\sigma^m_{ab} = \frac{1}{ig}\Phi^{\text{us}}_{ab}, &\quad& \Phi^{\text{them}}_m\sigma^{m,ab} = \frac{1}{ig}\Phi^{\text{us},ab}, \\
&D^{\text{them}}_\mu\sigma^\mu_{\alpha\dot\beta} = iD^{\text{us}}_{\alpha\dot\beta}, &\quad& A^{\text{them}}_\mu\sigma^\mu_{\alpha\dot\beta} = \frac{i}{g}A^{\text{us}}_{\alpha\dot\beta}, &\quad& x^{\text{them}}_\mu = -4ix^{\text{us}}_\mu,
\end{aligned}
\tag{D.2}
$$

where our coordinate convention is chosen such that the momentum generator $P_\mu$ satisfy the standard conformal algebra (B.13).

The supercharges act on the field strength as

$$[Q_\alpha^m, F_{\beta\gamma}] = \epsilon_{\alpha(\beta} D_{\gamma)}{}^{\dot\delta} \overline{\Psi}_{\dot\delta}^m, \qquad\qquad [Q_\alpha^m, F_{\dot\beta\dot\gamma}] = -D_{\alpha(\dot\beta} \overline{\Psi}_{\dot\gamma)}^m,$$
$$[\overline{Q}_{\dot\alpha m}, F_{\dot\beta\dot\gamma}] = \epsilon_{\dot\alpha(\dot\beta} D^\delta{}_{\dot\gamma)} \Psi_{\delta m}, \qquad [\overline{Q}_{\dot\alpha m}, F_{\beta\gamma}] = -D_{\dot\alpha(\beta} \Psi_{\gamma)m}.$$
(D.5)

The equations of motion for the fermions are

$$D_{\beta\dot\alpha} \Psi_a^\beta = i[\Phi_{ab}, \overline{\Psi}_{\dot\alpha}^b], \quad D_{\alpha\dot\beta} \overline{\Psi}^{a\dot\beta} = i[\Phi^{ab}, \Psi_{\alpha b}].$$
(D.6)

For our purpose, we have only used the above commutators in the free theory, where we can ignore the commutator $[\Phi^{mp}, \Phi_{np}]$, and replace the covariant derivatives with the partial derivatives. It is easy to check that they are compatible with the superconformal algebra, i.e.

$$[Q_\alpha^m, \{\overline{Q}_{\dot\beta n}, X\}] + [\overline{Q}_{\dot\beta n}, \{Q_\alpha^m, X\}] = 2i\delta_n^m \partial_{\alpha\dot\beta} X,$$
$$[Q_\alpha^m, \{Q_\beta^n, X\}] + [Q_\beta^n, \{Q_\alpha^m, X\}] = 0,$$
(D.7)

up to the equations of motion, and with the identification of the momentum $P_{\alpha\dot\beta}$ and the derivative $\partial_{\alpha\dot\beta}$ as

$$P_{\alpha\dot\beta} = 2i\partial_{\alpha\dot\beta} = \frac{1}{2}i(\sigma^\mu)_{\alpha\dot\beta}\partial_\mu.$$
(D.8)

In our convention, the super-Yang-Mills action is

$$S = \frac{2}{g_{\text{YM}}^2} \int d^4x \, \text{Tr}\left[\frac{1}{4}F_{\mu\nu}F^{\mu\nu} - 2D^\mu\Phi^{mn}D_\mu\Phi_{mn} - 4[\Phi^{mn}, \Phi^{pq}][\Phi_{mn}, \Phi_{pq}]\right.$$
$$\left. - 64i\overline{\Psi}_{\dot\alpha}^m D^{\beta\dot\alpha}\Psi_{m\beta} + 32\Psi_m^\alpha[\Phi^{mn}, \Psi_{\alpha n}] + 32\overline{\Psi}^{m\dot\alpha}\left[\Phi_{mn}, \overline{\Psi}_{\dot\alpha}^n\right]\right]$$
$$+ \frac{\theta}{16\pi^2} \int d^4x \, \epsilon^{\mu\nu\rho\sigma}F_{\mu\nu}F_{\rho\sigma}.$$
(D.9)

It is the same as the (1.5), (1.14) in [24] by the map between the conventions given in Footnote 13.

# E Inner product of single BPS letters

## E.1 Inner product of matrix fields

Consider a scalar field $X$ valued in the adjoint representation of U($N$). The inner product between the matrix components of $X$ is diagonal

$$\langle X_J^I | X_K^L \rangle = \delta_K^J \delta_I^L.$$
(E.1)

When the scalar field $X$ is in the adjoint representation of SU($N$), the inner product is

$$\langle X_J^I | X_K^L \rangle = \delta_K^J \delta_I^L - \frac{1}{N}\delta_I^J \delta_K^L.$$
(E.2)

The inner product between the off-diagonal matrix elements of $X$ is diagonal, but between the diagonal matrix elements of $X$ is non-diagonal. We could consider linear combinations of the diagonal matrix elements of $X$ such that the inner product between them is diagonal. For example, for $N = 2$, we have

$$X_1^1 = \mathcal{X}_1,$$
$$X_2^2 = -\mathcal{X}_1.$$
(E.3)

For $N = 3$, we have

$$
\begin{aligned}
X_1^1 &= -\mathcal{X}_1 + \mathcal{X}_2 \,, \\
X_2^2 &= \mathcal{X}_1 + \mathcal{X}_2 \,, \\
X_3^3 &= -2\mathcal{X}_2 \,.
\end{aligned}
\tag{E.4}
$$

For $N = 4$, we have

$$
\begin{aligned}
X_1^1 &= -\mathcal{X}_1 - \mathcal{X}_2 + \mathcal{X}_3 \,, \\
X_2^2 &= 2\mathcal{X}_2 + \mathcal{X}_3 \,, \\
X_3^3 &= \mathcal{X}_1 - \mathcal{X}_2 + \mathcal{X}_3 \,, \\
X_4^4 &= -3\mathcal{X}_3 \,.
\end{aligned}
\tag{E.5}
$$

The inner product matrices of $\mathcal{X}_i$ are diagonal matrices with the diagonal components $\langle \mathcal{X}_1 | \mathcal{X}_1 \rangle = \frac{1}{2}$, $\langle \mathcal{X}_2 | \mathcal{X}_2 \rangle = \frac{1}{6}$, $\langle \mathcal{X}_3 | \mathcal{X}_3 \rangle = \frac{1}{12}$.

### E.2 Inner products of BPS letters without derivatives

Let us consider BPS letters (16) without covariant derivatives, which corresponds to the conformal primary states

$$
\left| \phi^i \right\rangle, \quad |\psi_i\rangle, \quad |\lambda_{\dot{\alpha}}\rangle, \quad |f\rangle.
\tag{E.6}
$$

We presently compute the inner products of the fundamental fields (D.1) in the free theory. Using the action (D.9), we compute the two-point function of the scalar $\Phi_{mn}$,

$$
\langle \Phi^{mn}(x)\Phi_{pq}(0) \rangle = \frac{g_{\text{YM}}^2}{32\pi^2} \frac{\delta_{[p}^m \delta_{q]}^n}{|x|^2} \,.
\tag{E.7}
$$

The inner product of $|\Phi_{mn}\rangle$ is given by the limit

$$
\langle \Phi^{mn} | \Phi_{pq} \rangle = \lim_{x \to \infty} |x|^2 \langle \Phi^{mn}(x)\Phi_{pq}(0) \rangle = \frac{g_{\text{YM}}^2}{32\pi^2} \delta_{[p}^m \delta_{q]}^n \,.
\tag{E.8}
$$

The fermions $\Psi_{\alpha m}$, $\overline{\Psi}_{\dot{\alpha}}^m$ and the field strength $F_{\alpha\beta}$ are related to the scalar by

$$
\begin{aligned}
|\Psi_{\alpha m}\rangle &= \frac{1}{3} Q_\alpha^n |\Phi_{nm}\rangle \,, \quad |\overline{\Psi}_{\dot{\alpha}}^m\rangle = -\frac{1}{3}\overline{Q}_{\dot{\beta}n}|\Phi^{nm}\rangle \,, \\
|F_{\alpha\beta}\rangle &= \frac{i}{8} Q_{(\alpha}^m |\Psi_{\beta)m}\rangle = -\frac{i}{24} Q_\alpha^m Q_\beta^n |\Phi_{mn}\rangle \,.
\end{aligned}
\tag{E.9}
$$

Their inner product can be computed by using the superconformal algebra given in Appendix B, and the fact that the scalars $\Phi_{mn}$'s are superconformal primaries, i.e.

$$
S_m^\alpha |\Phi_{nq}\rangle = 0 = \overline{S}_{\dot{\alpha}}^m |\Phi_{nq}\rangle \,.
\tag{E.10}
$$

We compute the inner product of the fermions and field strength,[14]

$$
\langle \Psi^{\alpha p} | \Psi_{\beta q} \rangle = \frac{1}{9} \langle \Phi^{mp} | \{S_m^\alpha, Q_\beta^n\} |\Phi_{nq}\rangle = \frac{g_{\text{YM}}^2}{64\pi^2} \delta_\beta^\alpha \delta_q^p \,,
$$

---

[14]The inner products can also be obtained from the two-point functions

$$
\begin{aligned}
&\langle \Psi^{\alpha m} | \Psi_{\beta n} \rangle = \lim_{x \to \infty} |x|^2 x_\mu (\sigma^\mu)^{\alpha \dot{\gamma}} \langle \overline{\Psi}_{\dot{\gamma}}^m(x)\Psi_{\beta n}(0) \rangle \,, \quad \langle \overline{\Psi}_m^{\dot{\alpha}} | \overline{\Psi}_{\dot{\beta}}^n \rangle = \lim_{x \to \infty} |x|^2 x_\mu (\sigma^\mu)^{\gamma \dot{\alpha}} \langle \Psi_{\gamma m}(x)\overline{\Psi}_{\dot{\beta}}^n(0) \rangle \,, \\
&\langle F^{\alpha\beta} | F_{\gamma\delta} \rangle = \lim_{x \to \infty} |x|^2 x_\mu (\sigma^\mu)^{\alpha\dot{\alpha}} x_\nu (\sigma^\nu)^{\beta\dot{\beta}} \langle F_{\dot{\alpha}\dot{\beta}}(x)F_{\gamma\delta}(0) \rangle \,.
\end{aligned}
\tag{E.11}
$$

$$\langle \overline{\Psi}_p^{\dot\alpha} | \overline{\Psi}_{\dot\beta}^q \rangle = \frac{1}{9} \langle \Phi_{mp} | \{ \overline{S}^{\dot\alpha m}, \overline{Q}_{\dot\beta n} \} | \Phi^{nq} \rangle = \frac{g_{\text{YM}}^2}{64\pi^2} \delta_{\dot\beta}^{\dot\alpha} \delta_p^q \,,$$

$$\langle F^{\alpha\beta} | F_{\gamma\delta} \rangle = \langle \Psi^{\beta p} | S_p^\alpha Q_\gamma^m | \Psi_{\delta m} \rangle \tag{E.12}$$

$$= \frac{1}{64} \langle \Psi^{\beta p} | \{ S_p^\alpha, Q_\gamma^m \} | \Psi_{\delta m} \rangle - \frac{1}{64} \langle \Psi^{\beta p} | Q_\gamma^m S_p^\alpha | \Psi_{\delta m} \rangle$$

$$= \frac{g_{\text{YM}}^2}{32\pi^2} \left( \frac{1}{8} \delta_\gamma^\alpha \delta_\delta^\beta + \frac{1}{32} \delta_\delta^\alpha \delta_\gamma^\beta \right) - \frac{1}{64} \langle \Psi^{\beta p} | Q_\gamma^m S_p^\alpha | \Psi_{\delta m} \rangle$$

$$= \frac{g_{\text{YM}}^2}{256\pi^2} \delta_{(\gamma}^\alpha \delta_{\delta)}^\beta \,.$$

Specializing the above results to the BPS letters, we find

$$\langle \phi_i | \phi^j \rangle = \frac{g_{\text{YM}}^2}{64\pi^2} \delta_i^j \,, \quad \langle \psi^i | \psi_j \rangle = \frac{g_{\text{YM}}^2}{64\pi^2} \delta_j^i \,, \quad \langle \lambda^{\dot\alpha} | \lambda_{\dot\beta} \rangle = \frac{g_{\text{YM}}^2}{64\pi^2} \delta_{\dot\beta}^{\dot\alpha} \,, \quad \langle f | f \rangle = \frac{g_{\text{YM}}^2}{256\pi^2} \,. \tag{E.13}$$

### E.3 Inner products of BPS letters with derivatives

Let us consider BPS letters with covariant derivatives. In free theory, the covariant derivatives become partial derivatives and are related to the momentum generator $P_{\alpha\dot\beta}$ as

$$D_{\alpha\dot\beta} = \partial_{\alpha\dot\beta} = \frac{1}{2i} P_{\alpha\dot\beta} \,. \tag{E.14}$$

Hence, these BPS letters are descendent of the conformal primary letters (E.6), and take the form as

$$P_{\dot\alpha_1} \cdots P_{\dot\alpha_n} | X \rangle \,, \tag{E.15}$$

for $|X\rangle = |\phi^i\rangle$, $|\psi_i\rangle$, $|\lambda_{\dot\alpha}\rangle$, or $|f\rangle$. Their inner products are

$$\langle X | K^{\dot\alpha_n} \cdots K^{\dot\alpha_1} P_{\dot\beta_1} \cdots P_{\dot\beta_n} | X \rangle \,, \tag{E.16}$$

which are reduced to the inner product $\langle X | X \rangle$ by the commutators in the $\mathfrak{su}(1,2)$ subalgebra given in Appendix C. In particular, the commutators (C.7) imply

$$K^{\dot-} P_{\dot-}^n P_{\dot+}^m = n(D + m + n - 1) P_{\dot-}^{n-1} P_{\dot+}^m - n P_{\dot-}^{n-1} P_{\dot+}^m M^{\dot-}{}_{\dot-}$$
$$- m P_{\dot-}^n P_{\dot+}^{m-1} M^{\dot-}{}_{\dot+} + P_{\dot-}^n P_{\dot+}^m K^{\dot-} \,,$$
$$K^{\dot+} P_{\dot-}^n P_{\dot+}^m = m(D + m + n - 1) P_{\dot-}^n P_{\dot+}^{m-1} - m P_{\dot-}^n P_{\dot+}^{m-1} M^{\dot+}{}_{\dot+}$$
$$- n P_{\dot-}^{n-1} P_{\dot+}^m M^{\dot+}{}_{\dot-} + P_{\dot-}^n P_{\dot+}^m K^{\dot+} \,. \tag{E.17}$$

The $M^{\dot\alpha}{}_{\dot\beta}$ action on the conformal primary states as,

$$M^{\dot\alpha}{}_{\dot\beta} | \phi^i \rangle = 0 \,, \qquad\qquad M^{\dot\alpha}{}_{\dot\beta} | \psi_i \rangle = -\frac{1}{2} \delta_{\dot\beta}^{\dot\alpha} | \psi_i \rangle \,,$$
$$M^{\dot\alpha}{}_{\dot\beta} | \lambda_{\dot\gamma} \rangle = -\delta_{\dot\gamma}^{\dot\alpha} | \lambda_{\dot\beta} \rangle + \frac{1}{2} \delta_{\dot\beta}^{\dot\alpha} | \lambda_{\dot\gamma} \rangle \,, \quad M^{\dot\alpha}{}_{\dot\beta} | f \rangle = -\delta_{\dot\beta}^{\dot\alpha} | f \rangle \,. \tag{E.18}$$

Now, let us define

$$F_X(m, n) \equiv \langle X | (K^{\dot+})^m (K^{\dot-})^n P_{\dot-}^n P_{\dot+}^m | X \rangle \,, \tag{E.19}$$

for $X = \phi^i$, $\psi_i$, $f$, and

$$F_{+,+}(m, n) \equiv \langle \lambda^{\dot+} | (K^{\dot+})^m (K^{\dot-})^n P_{\dot-}^n P_{\dot+}^m | \lambda_{\dot+} \rangle \,,$$
$$F_{-,-}(m, n) \equiv \langle \lambda^{\dot-} | (K^{\dot+})^m (K^{\dot-})^n P_{\dot-}^n P_{\dot+}^m | \lambda_{\dot-} \rangle \,,$$
$$F_{+,-}(m, n) \equiv \langle \lambda^{\dot+} | (K^{\dot+})^{m-1} (K^{\dot-})^n P_{\dot-}^{n-1} P_{\dot+}^m | \lambda_{\dot-} \rangle \,,$$
$$F_{-,+}(m, n) \equiv \langle \lambda^{\dot-} | (K^{\dot+})^m (K^{\dot-})^{n-1} P_{\dot-}^n P_{\dot+}^{m-1} | \lambda_{\dot+} \rangle \,. \tag{E.20}$$

From (E.17), we find

$$
\begin{aligned}
F_X(m,n) &= n(\Delta_X + m + n - 1 + J_L)F_X(m, n-1), \\
F_X(m,n) &= m(\Delta_X + m + n - 1 + J_L)F_X(m-1, n),
\end{aligned}
\tag{E.21}
$$

where $\Delta_X$ and $J_X$ are the eigenvalues of $D$ and $(J_L)_-^-$. We also find

$$
\begin{aligned}
F_{+,+}(m,n) &= n(m+n)F_{+,+}(m, n-1), \\
F_{+,+}(m,n) &= m(m+n+1)F_{+,+}(m-1, n) + nF_{+,-}(m, n), \\
F_{-,-}(m,n) &= n(m+n+1)F_{-,-}(m, n-1) + mF_{-,+}(m, n), \\
F_{-,-}(m,n) &= m(m+n)F_{-,-}(m-1, n),
\end{aligned}
\tag{E.22}
$$

where we have used (E.18). We also have

$$
\begin{aligned}
F_{+,-}(m,n) &= (n-1)(m+n)F_{+,-}(m, n-1) + mF_{+,+}(m-1, n-1), \\
F_{+,-}(m,n) &= m(m+n)F_{+,-}(m-1, n), \\
F_{-,+}(m,n) &= n(m+n-1)F_{-,+}(m, n-1), \\
F_{-,+}(m,n) &= (m-1)(m+n)F_{-,+}(m-1, n) + nF_{-,-}(m-1, n-1).
\end{aligned}
\tag{E.23}
$$

We solve these recurrence relations. The solutions are

$$
F_X(m,n) = \frac{\Gamma(m+1)\Gamma(n+1)\Gamma(\Delta_X + J_{L,X} + m + n)}{\Gamma(\Delta_X + J_{L,X})}\langle \bar{X}|X\rangle,
\tag{E.24}
$$

and

$$
\begin{aligned}
F_{+,+}(m,n) &= \Gamma(m+2)\Gamma(n+1)\Gamma(m+n+1)\langle \bar{\lambda}^{\dot{+}}|\lambda_{\dot{+}}\rangle, \\
F_{-,-}(m,n) &= \Gamma(m+1)\Gamma(n+2)\Gamma(m+n+1)\langle \bar{\lambda}^{\dot{-}}|\lambda_{\dot{-}}\rangle, \\
F_{+,-}(m,n) &= \Gamma(m+1)\Gamma(n+1)\Gamma(m+n)\langle \bar{\lambda}^{\dot{+}}|\lambda_{\dot{+}}\rangle, \\
F_{-,+}(m,n) &= \Gamma(m+1)\Gamma(n+1)\Gamma(m+n)\langle \bar{\lambda}^{\dot{-}}|\lambda_{\dot{-}}\rangle.
\end{aligned}
\tag{E.25}
$$

Let us summarize these inner products by using the BPS superfield.

$$
\phi^i = \frac{i}{2}\partial_{\theta_i}\Psi, \qquad \psi_i = -\frac{i}{4}\epsilon_{ijk}\partial_{\theta_j}\partial_{\theta_k}\Psi, \qquad \lambda_{\dot{\alpha}} = i\partial_{z^{\dot{\alpha}}}\Psi, \qquad f = -\frac{i}{4}\partial_{\theta_1}\partial_{\theta_2}\partial_{\theta_3}\Psi, \quad
\tag{E.26}
$$

First, using the relation

$$
\partial_{z^+}^m\partial_{z^-}^n\Psi(\mathcal{Z})\big|_{\mathcal{Z}=0} = -i\frac{m(\frac{1}{2i}P_+)^{m-1}(\frac{1}{2i}P_-)^n\lambda_+ + n(\frac{1}{2i}P_+)^m(\frac{1}{2i}P_-^{n-1})\lambda_-}{m+n},
\tag{E.27}
$$

we find

$$
\begin{aligned}
&\left\langle \left(\partial_{z^+}^m\partial_{z^-}^n\Psi\right)^\dagger \big| \partial_{z^+}^m\partial_{z^-}^n\Psi\right\rangle \\
&= \frac{4^{1-m-n}}{(m+n)^2}\Big[m^2 F_{\bar{\lambda}^{\dot{+}},\lambda_{\dot{+}}}(m-1,n) + n^2 F_{\bar{\lambda}^{\dot{-}},\lambda_{\dot{-}}}(m,n-1) + mn F_{\bar{\lambda}^{\dot{+}},\lambda_{\dot{-}}}(m,n) + mn F_{\bar{\lambda}^{\dot{-}},\lambda_{\dot{+}}}(m,n)\Big] \\
&= 4^{1-m-n}\Gamma(m+1)\Gamma(n+1)\Gamma(m+n)\langle \bar{\lambda}^{\dot{+}}|\lambda_{\dot{+}}\rangle.
\end{aligned}
\tag{E.28}
$$

It is straightforward to find the following formulae

$$
\begin{aligned}
\left\langle \left(\partial_{z^+}^m\partial_{z^-}^n\partial_{\theta_1}\Psi\right)^\dagger \big| \partial_{z^+}^m\partial_{z^-}^n\partial_{\theta_1}\Psi\right\rangle &= 4^{1-m-n}\Gamma(m+1)\Gamma(n+1)\Gamma(m+n+1)\langle \bar{\phi}_1|\phi^1\rangle, \\
\left\langle \left(\partial_{z^+}^m\partial_{z^-}^n\partial_{\theta_1}\partial_{\theta_2}\Psi\right)^\dagger \big| \partial_{z^+}^m\partial_{z^-}^n\partial_{\theta_1}\partial_{\theta_2}\Psi\right\rangle &= 4^{1-m-n}\Gamma(m+1)\Gamma(n+1)\Gamma(m+n+2)\langle \bar{\psi}^3|\psi_3\rangle, \\
\left\langle \left(\partial_{z^+}^m\partial_{z^-}^n\partial_{\theta_1}\partial_{\theta_2}\partial_{\theta_3}\Psi\right)^\dagger \big| \partial_{z^+}^m\partial_{z^-}^n\partial_{\theta_1}\partial_{\theta_2}\partial_{\theta_3}\Psi\right\rangle &= \frac{1}{2}4^{2-m-n}\Gamma(m+1)\Gamma(n+1)\Gamma(m+n+3)\langle \bar{f}|f\rangle,
\end{aligned}
\tag{E.29}
$$

and similar results by permuting the $\theta_1$, $\theta_2$, $\theta_3$. With the inner products of the primary states (E.13), we find

$$\left\langle \partial_{z^+}^{a_1} \partial_{z^-}^{a_2} \partial_{\theta_1}^{a_3} \partial_{\theta_2}^{a_4} \partial_{\theta_3}^{a_5} \Psi \middle| \partial_{z^+}^{a_1} \partial_{z^-}^{a_2} \partial_{\theta_1}^{a_3} \partial_{\theta_2}^{a_4} \partial_{\theta_3}^{a_5} \Psi \right\rangle$$

$$= \frac{g_{\text{YM}}^2}{2^{4+2a_1+2a_2} \pi^2} \Gamma(a_1+1)\Gamma(a_2+1)\Gamma(a_1+a_2+a_3+a_4+a_5). \qquad \text{(E.30)}$$

## F  Alternative expression for the smallest black hole operator

In [15], an expression for the smallest black hole operator of [14] was presented on paper. This appendix translates the superfield notation of Section 5.2 to the notation of [15] up to normalization. Writing $\vec{\sigma}$ as a vector of Pauli matrices,[15] we define $\vec{\phi}^i$, $\vec{\psi}_i$ and $\vec{f}$ through

$$\vec{\phi}^i \cdot \vec{\sigma} = \frac{i}{2}\partial_{\theta_i}\Psi, \qquad \vec{\psi}_i \cdot \vec{\sigma} = -\frac{i}{4}\epsilon_{ijk}\partial_{\theta_j}\partial_{\theta_k}\Psi, \qquad \vec{f} \cdot \vec{\sigma} = -\frac{i}{4}\partial_{\theta_1}\partial_{\theta_2}\partial_{\theta_3}\Psi. \qquad \text{(F.1)}$$

Then

$$O'_1 = 4096(\vec{f} \cdot \vec{f})(\vec{f} \cdot \vec{\phi}^1)(\vec{\phi}^2 \cdot \vec{\phi}^3) + \text{cyclic},$$
$$O'_2 = 2048(\vec{f} \cdot \vec{f})(\vec{\phi}^1 \cdot \vec{\phi}^2)(\vec{\psi}_1 \cdot \vec{\psi}_2) + \text{cyclic},$$
$$O'_3 = 2048(\vec{f} \cdot \vec{f})(\vec{\phi}^1 \cdot \vec{\psi}_1)(\vec{\phi}^2 \cdot \vec{\psi}_2) + \text{cyclic},$$
$$O'_4 = 2048(\vec{f} \cdot \vec{\phi}^1)(\vec{f} \cdot \vec{\phi}^2)(\vec{\psi}_1 \cdot \vec{\psi}_2) + \text{cyclic},$$
$$O'_5 = 2048(\vec{f} \cdot \vec{\psi}_1)(\vec{f} \cdot \vec{\psi}_2)(\vec{\phi}^1 \cdot \vec{\phi}^2) + \text{cyclic},$$
$$O'_6 = 2048(\vec{f} \cdot \vec{\phi}^1)(\vec{f} \cdot \vec{\psi}_1)(\vec{\phi}^2 \cdot \vec{\psi}_2) + \text{cyclic},$$
$$O'_7 = 2048(\vec{f} \cdot \vec{\phi}^2)(\vec{f} \cdot \vec{\psi}_1)(\vec{\phi}^1 \cdot \vec{\psi}_2) + \text{cyclic},$$
$$O'_8 = 2048(\vec{f} \cdot \vec{\phi}^1)(\vec{f} \cdot \vec{\psi}_2)(\vec{\phi}^2 \cdot \vec{\psi}_1) + \text{cyclic},$$
$$O'_9 = 2048(\vec{f} \cdot \vec{\phi}^2)(\vec{f} \cdot \vec{\psi}_2)(\vec{\phi}^1 \cdot \vec{\psi}_1) + \text{cyclic},$$
$$O'_{10} = 2048(\vec{f} \cdot \vec{f})(\vec{\phi}^1 \cdot \vec{\psi}_2)(\vec{\phi}^2 \cdot \vec{\psi}_1) + \text{cyclic}, \qquad \text{(F.2)}$$
$$O'_{11} = -1024(\vec{f} \cdot \vec{\phi}^1)(\vec{\psi}_1 \cdot \vec{\psi}_2)(\vec{\psi}_1 \cdot \vec{\psi}_3) + \text{cyclic},$$
$$O'_{12} = -1024(\vec{f} \cdot \vec{\psi}_2)(\vec{\psi}_1 \cdot \vec{\phi}^1)(\vec{\psi}_1 \cdot \vec{\psi}_3) + \text{cyclic},$$
$$O'_{13} = -1024(\vec{f} \cdot \vec{\psi}_3)(\vec{\psi}_1 \cdot \vec{\psi}_2)(\vec{\psi}_1 \cdot \vec{\phi}^1) + \text{cyclic},$$
$$O'_{14} = -1024(\vec{f} \cdot \vec{\psi}_1)(\vec{\phi}^1 \cdot \vec{\psi}_2)(\vec{\psi}_1 \cdot \vec{\psi}_3) + \text{cyclic},$$
$$O'_{15} = -2048(\vec{f} \cdot \vec{\phi}^1)(\vec{f} \cdot \vec{\psi}_1)(\vec{\phi}^1 \cdot \vec{\psi}_1) + \text{cyclic},$$
$$O'_{16} = -1024(\vec{f} \cdot \vec{\psi}_1)(\vec{\psi}_2 \cdot \vec{\psi}_3)(\vec{\phi}^1 \cdot \vec{\psi}_1) + \text{cyclic},$$
$$O'_{17} = -512(\vec{\psi}_1 \cdot \vec{\psi}_2)(\vec{\psi}_2 \cdot \vec{\psi}_3)(\vec{\psi}_3 \cdot \vec{\psi}_1),$$

and

$$O = -1024(\vec{\psi}_1 \cdot \vec{\phi}^1 - \vec{\psi}_2 \cdot \vec{\phi}^2)(\vec{\psi}_3 \cdot \vec{\phi}^1)\vec{\psi}_2 \cdot (\vec{\psi}_1 \times \vec{\psi}_1) + \text{cyclic}. \qquad \text{(F.3)}$$

---

[15]Note that in terms of the $\sigma$-matrices defined in (A.1), we have $\vec{\sigma} = (-i\sigma^1, -i\sigma^2, -i\sigma^3)$.

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
