# Peer review of "Decoding stringy near-supersymmetric black holes"

_SciPost Physics, doi:SciPost Phys. 16, 109 (2024)_

## Round 1 · Referee Report · Anonymous (Referee 3) · 2024-2-18

Report

I thank the authors for the clarifications contained in their reply. I would like to add a couple of comments on some of the changes made to the new version of the article.

  1. We added Section 3.4.

The argument at the beginning of Section 3.4: "The anomalous dimensions of multi-trace operators are given by the sum of those of the single-trace constituents." is true only if one takes the large N limit keeping the dimension of the operator, or equivalently the number traces (denoted by $n$ in (3.28)), fixed. The limit that is relevant for comparison with a macroscopic black hole in the supergravity approximation is the so-called "heavy operator" limit in which $n/N^2$ is kept fixed when $N$ is taken to infinity. In this limit the anomalous dimensions resulting from the interaction between the various traces generically remain finite (unless the multi-trace operator is BPS also in the interacting theory, of course), and one cannot just focus on the single trace operators to study the cohomology of the interacting theory. This fact is well understood also from the gravity point of view, both when one considers black hole geometries or horizon-less geometries dual to the multi-graviton states.

  1. We thank the referee for pointing this out. The statement at the end of Section 5.1 was misleading and partially incorrect. We have removed that statement.

Actually the statement I was pointing out is: "In the non-BPS case, there is no known method to distinguish multi-gravitons from potential black holes, thus putting an asterisk on how much our data reflects stringy black hole physics.7" and it is still present in the new version. I think it would be useful to clarify why the authors make that claim and also how this is compatible with the reference mentioned in footnote 7.

---

## Round 1 · Referee Report · Anonymous (Referee 2) · 2024-3-3

Report

I thank the authors for the clarifications for which I asked.
The manuscript contains a number of inspiring results and I recommend it for the publication in SciPost.

---

## Round 1 · List of Changes

1. We added footnote 3 the explain the orders of $g_YM$ in the inner product and the $Q$-action.

  2. We added footnote 6 and a sentence at the end of the caption of Figure 3 for clarification.

  3. We thank the referee for pointing out this error. The matrix $T_Y$ was computed in the previous section (Section 3.2). We have deleted that sentence.

  4. We added Section 3.4.

  5. We added an additional column “# BPS operators” to Table 1.

  6. We have added some discussions at the end of section 3.3 explaining how one distinguishes whether a BPS operator is a (multi-)graviton operator.

  7. We thank the referee for pointing this out. The statement at the end of Section 5.1 was misleading and partially incorrect. We have removed that statement. In the classically-BPS sector with respect to the supercharge $Q^4_-$, the non-BPS operators are always non-graviton states. They could be massive string states or bound states of massive strings and gravitons.

  8. We thank the referee for pointing out the typo in eq. (3.6). It has been corrected.

---

## Round 2 · List of Changes

1. We added footnote 4 specifying the regime of our discussion.

  2. We added footnote 5 showing that the strict large N limit is well-defined in the space of BPS words.

  3. We thank the referee for pointing it out. We rephrased our statement in Sec. 5.1.

---

## Editorial Decision

published